



# Easy Volcanic Aerosol (EVA v1.0): An idealized forcing generator for climate simulations

Matthew Toohey[1,2], Bjorn Stevens[1], Hauke Schmidt[1], Claudia Timmreck[1]

[1] Max Planck Institute for Meteorology, Hamburg, 20146, Germany

5    [2] GEOMAR, Helmholtz Centre for Ocean Research Kiel, 24105, Germany

*Correspondence to*: Matthew Toohey (mtoohey@geomar.de)

**Abstract.** The Easy Volcanic Aerosol (EVA) forcing generator produces stratospheric aerosol optical properties as a function of time, latitude, height and wavelength for a given input list of volcanic eruption attributes. EVA is based on a

10    parameterized three-box model of stratospheric transport, and simple scaling relationships used to derive mid-visible (550 nm) aerosol optical depth and aerosol effective radius from stratospheric sulfate mass. Pre-calculated look up tables computed from Mie theory are used to produce wavelength dependent aerosol extinction, single scattering albedo and scattering asymmetry factor values. The structural form of EVA, and the tuning of its parameters, are chosen to produce best agreement with the satellite-based reconstruction of stratospheric aerosol properties following the 1991 Pinatubo eruption,

15    and with prior millennial-time scale forcing reconstructions including the 1815 eruption of Tambora. EVA can be used to produce volcanic forcing for climate models which is based on recent observations and physical understanding, but internally self-consistent over any time-scale of choice. In addition, EVA is constructed so as to allow for easy modification of different aspects of aerosol properties, in order to be used in model experiments to help advance understanding of what aspects of the volcanic aerosol are important for the climate system.



## 1 Introduction

Radiative forcing by variations of stratospheric sulfate aerosol from volcanic eruptions is one of the strongest drivers of natural climate variability (Crowley, 2000; Schurer et al., 2013). To reproduce the radiative forcing of past volcanic eruptions, and thereby the related climate variability, climate model simulations require estimates of the optical properties of

volcanic stratospheric aerosols. Prognostic stratospheric aerosol schemes are available, however, such schemes are computationally expensive, and many of the processes underlying them are still not well understood. For these reasons, transient simulations such as historical or millennium simulations usually rely on prescriptive volcanic forcing reconstructions (where "forcing" hereafter refers not specifically to "radiative forcing", but rather to any external driver of climate variability prescribed in climate model simulations). Different volcanic aerosol forcing sets are currently available,

which use different data sources, different methodologies for combining data sources, and provide different—often incomplete—representations of aerosol properties needed for the radiative calculations of climate models.

The response of the Earth system to volcanic forcing simulated by climate models has been seen to be unrealistic in a number of prior studies. Stratospheric heating, due to the absorption of infrared radiation by volcanic aerosols appears to be overestimated in some models (Driscoll et al., 2012). Tropospheric cooling, while relatively realistically simulated for recent

eruptions (Santer et al., 2014), appears to be too strong in model simulations for a number of large past eruptions, most noticeably for the eruptions of Tambora in 1815 (Brohan et al., 2012) and Samalas in 1257 (Stoffel et al., 2015). Post-volcanic anomalies of atmospheric circulation, inferred from observations, are not robustly simulated by models with prescribed forcing, either at the surface (Driscoll et al., 2012), or in the stratosphere (Charlton-Perez et al., 2013). There is also a large degree of inter-model spread in the temperature response to volcanic eruptions, and even in the radiative

anomalies created by prescribed volcanic forcing (Zanchettin et al., 2015). Because there are differences in the forcing data sets used, some of which may result from differences in their implementation, it remains unclear to what degree inter-model spread in response to volcanic forcing is attributable to differences in the climate models (i.e., model uncertainty) or differences in forcing (or its implementation).

In general, to isolate model uncertainty in the response to external forcings, it is desirable to have a single forcing

implementation strategy, which can be applied consistently in different models. To test the sensitivity to different aspects of the forcing, and gain understanding as to what aspects of the forcing are important for the climate response, it is further desirable to have a forcing strategy which is flexible enough to be used in sensitivity studies. Such motivations inspired the "Easy Aerosol" approach to prescribed tropospheric aerosols (Voigt et al., 2014), wherein the spatial structure of tropospheric aerosols was defined by simple analytical functions in latitude and longitude. Building upon the Easy Aerosol

approach, the MACv2-SP module provides relatively realistic representations of anthropogenic aerosol plumes, with a large degree of flexibility and utility for idealized studies (Stevens et al., in preparation).



We present here a description of the Easy Volcanic Aerosol (EVA) forcing generator for use in climate model simulations. EVA provides models with the full optical properties of volcanic aerosols in terms of wavelength dependent aerosol extinction, single scattering albedo, and asymmetry factor, given an input list of eruption locations, dates, and estimated stratospheric sulfur injections. The spatio-temporal structure of the prescribed forcing aims to strike a balance between being realistic (compared to modern observations), and generic, therefore producing consistent representations of eruptions over arbitrary time periods. The underlying parameterization is also readily modifiable, lending itself naturally to idealized sensitivity studies. EVA is comprised of a FORTRAN module that can be called directly by climate models, or can be used offline to produce forcing files which a model reads upon integration.

EVA builds directly upon the methods and results of previous volcanic aerosol forcing reconstructions, which are briefly described in Sec. 2. The EVA approach is detailed in Sec. 3, and a brief comparison with other reconstructions is included in Sec. 4. A summary of EVA, and outlook to potential uses and future versions is included in Sec 5.

## 2 Existing data sets and approaches

There is a large body of work concerning reconstruction of the radiative forcing of stratospheric aerosol from volcanic eruptions. We briefly introduce below reconstructions which have been most often used in recent climate model simulations, and which are integral in the construction of EVA.

### 2.1 Sato/GISS

The Sato/GISS forcing data set (Sato et al., 1993, with updates: http://data.giss.nasa.gov/modelforce/strataer/) provides stratospheric aerosol optical depth at 550 nm ($AOD_{550}$) and aerosol effective radius ($r_{eff}$) as a function of latitude, height and time for the period 1850-present. It is based on a mixture of satellite observations, ground-based optical measurements, and volcanological evidence. From 2001-present, the reconstruction is based exclusively on Optical Spectrograph and InfraRed Imager System (OSIRIS) satellite measurements (Bourassa et al., 2008). For eruptions before the satellite era, the spatial structure of the $AOD_{550}$ is approximated, based on roughly scaled versions of observed eruptions, or global or hemispheric means. Aerosol effective radius is given as an empirical function of $AOD_{550}$, based on retrievals of AOD and $r_{eff}$ following the Pinatubo eruption.

### 2.2 Ammann et al. 2003

The Ammann et al., (2003) reconstruction provides $AOD_{550}$ as a function of latitude, height and time for the period 1890-1999. It is based on estimates of the mass of $SO_2$ injected into the stratosphere, $M_{SO2}$, from past eruptions. The spatial structure of the $AOD_{550}$ is produced by a parameterized stratospheric transport routine, representing the growth of AOD, its decay due to cross-tropopause transport, and transport from tropics to high latitudes. Compared to the Sato/GISS





reconstruction, this method provides a consistent representation of volcanic forcing for all eruptions, however, it clearly simplifies the volcanic cloud evolutions compared to the observed evolutions of eruptions like Pinatubo.

### 2.3 Gao et al., 2008

The Gao et al., (2008) reconstruction provides stratospheric sulfate aerosol mass as a function of latitude, height and time for

the period 500-2000, as well as estimates of $SO_2$ injection by individual volcanic events over the same period. $SO_2$ injections are based on ice core records of sulfate flux from Greenland and Antarctica. Using different scaling factors for tropical and extratropical eruptions, based on nuclear bomb tests and modeling studies, ice core-derived sulfate surface deposition rates are scaled to $SO_2$ injections (Gao et al., 2007). Sulfate aerosol mass timeseries, as a function of latitude and height, were produced using a modified version of the parameterized stratospheric aerosol transport scheme of Grieser and Schönwiese,

(1999). Conversion of sulfate aerosol mass to radiative properties was left to the implementation of the model, but a linear scaling of aerosol mass to AOD is a commonly used assumption (Schmidt et al., 2011).

### 2.4 Stenchikov

The "Stenchikov" reconstruction provides monthly mean zonal averages of stratospheric aerosol extinction, single scattering albedo, and asymmetry factor as a function of time, pressure, and wavelength, for the historic period from 1850 to 1999

(Driscoll et al., 2012; Schmidt et al., 2013). This data set was used by the Max Planck Institute Earth System Model and the Geophysical Fluid Dynamics Laboratory's Coupled Model for historical simulations as part of the Coupled Model Intercomparison Project's fifth phase (CMIP5). It is an extended version of the Pinatubo aerosol data set developed by Stenchikov et al. (1998) on the basis of satellite measurements of aerosol extinction and $r_{eff}$ after the Pinatubo eruption. For earlier eruptions, it is based on the Sato/GISS AOD550 and $r_{eff}$ reconstruction. Stratospheric background aerosols are

ignored, and only sulfate aerosols arising from volcanic eruptions are accounted for.

### 2.5 Crowley & Unterman 2013

The Crowley and Unterman (2013, hereafter CU13) reconstruction provides $AOD_{550}$ and aerosol effective radius ($r_{eff}$) in four equal-area latitude bands, for the time period 800-2000. It is based on ice core records of sulfate flux from Greenland to Antarctica. Ice core sulfate fluxes are scaled to peak $AOD_{550}$: for eruptions of Pinatubo magnitude and smaller, this scaling is

linear and based on the ratio of satellite-observed SH AOD and the flux of sulfate to Antarctica. For stronger eruptions (like Tambora), CU13 introduced the use of a nonlinear scaling, with $AOD_{550}$ proportion to the 2/3 power of sulfate flux. AOD timeseries were constructed based on assumptions of linear increase to peak value, a plateau, followed by exponential decay with a 12-month timescale. Like in the Sato/GISS reconstruction, effective radius is prescribed as a simple empirical function of AOD.



## 2.6 CCMI/SAGE_4λ

The aerosol forcing constructed for use in the Chemistry-climate model initiative (CCMI) (Eyring and Lamarque, 2013) provides aerosol optical properties aerosol extinction (EXT), single scattering albedo (SSA) and scattering asymmetry factor (ASY) as a function of wavelength, latitude, height and time for the period 1960-present. It also provides internally

consistent estimates of aerosol surface area density (SAD) necessary for stratospheric chemistry simulations. The cornerstone of the CCMI forcing set is the four-wave-length SAGE II extinction data (SAGE_4λ), retrieval version 7, which span the period 1985-2005. During the SAGE II period, the four-wavelength extinction data is used to derive estimates of aerosol effective radius, from which the properties EXT(λ), SSA(λ) and ASY(λ) are derived assuming a single lognormal particle size distribution (Arfeuille et al., 2013). For other time periods, aerosol radiative properties are estimated mainly

through single wavelength extinction retrievals from satellite instruments, and the observed correlation between mean aerosol radius and aerosol extinction during the SAGE II period. From 1979 to 1985 the reconstruction is based on single wavelength extinctions measured by the SAM II and SAGE I satellite instruments (Thomason and Peter, 2006). The 1960-1979 pre-satellite period has been constructed from SAGE-II background measurements in the late 1990s, superimposing the volcanic eruptions of Agung (1963) and Fuego (1974). These eruptions were calculated by means of the

AER 2-D aerosol model (Weisenstein et al., 1997), and the results were scaled by means of stellar and solar extinction data (Stothers, 2001). The 2006-2011 period is derived from CALIPSO 532nm backscatter data. Updates to the CCMI reconstruction will be used as historical forcing for the Coupled Model Intercomparison Project phase 6 (Thomason et al., 2016).

The observational data sets used in the CCMI reconstruction have data gaps, in particular when the atmosphere became

opaque directly after volcanic eruptions, which occurred mainly in lower tropical altitudes (below 16 km), and in the high latitude polar regions due to limited geographical sampling of the satellite instruments (Eyring and Lamarque, 2013). After the eruptions of El Chichón and Pinatubo, data gaps were filled by means of lidar ground station data and interpolation. Remaining data gaps were filled using a linear interpolation approach in altitude and latitude.

## 3. The EVA approach

### 3.1 Basis and input data

The construction of EVA relies extensively on observational constraints. Additionally, when observations cannot adequately constrain EVA parameterizations, EVA is constructed so as to produce reasonable agreement with prior forcing sets. Data sets that were instrumental in the EVA construction include:

- The estimate of total $SO_2$ injection by Pinatubo of 18 Tg (9 Tg S) from the Total Ozone Mapping Spectrometer
(TOMS) instrument (Guo, 2004).



- Aerosol extinction from the CCMI (Arfeuille et al., 2013). Here we have used the CCMI data provided at http://www.pa.op.dlr.de/CCMI/CCMI_SimulationsForcings.html, specifically the files produced for use in the ECHAM6 model. We focus primarily on the EXT at 550 nm (EXT550). AOD is computed based on the vertical integral of EXT above the climatological tropopause. Post-Pinatubo AOD and EXT anomalies are computed by subtracting an estimate of the "background" aerosol levels, based on the mean EXT field from the years 1999 and 2000.

- Estimates of aerosol effective radius provided in the Sato/GISS and CU13 data sets.

Input data, specifying the stratospheric sulfur injection properties for a number of volcanic eruptions were collected from a range of sources, summarized in Table 1. The ice core-derived sulfate aerosol loading estimates of Gao et al. (2008) were used to produce $SO_2$ injection estimates for the great Tambora eruption of 1815, as well as the eruptions of Agung (1963), Fuego (1974), and El Chichón (1982). For the 1991 eruption of Pinatubo, we use the estimated total $SO_2$ injection of 18 Tg (9 Tg S) from the Total Ozone Mapping Spectrometer (TOMS) instrument (Guo, 2004). Estimates of stratospheric $SO_2$ injection for a number of relatively smaller eruptions in the 2000's were taken from Brühl et al., (2015), based on the MIPAS $SO_2$ retrievals described by Höpfner et al., (2015).

## 3.2 Global mean AOD and effective radius

Aerosol optical depth in the mid-visible ($\lambda$=550 nm) is simulated by EVA making use of the assumption that it can be a simple function of the stratospheric sulfate mass, i.e.:

$$AOD_{550} = f(M_{SO4}) \tag{1}$$

(Stothers, 1984) introduced a simple linear scaling between global sulfate aerosol mass and global mean AOD. Such scalings have a physical foundation (e.g., Charlson et al., 1992) and have long been used to study tropospheric aerosols. A linear relationship between sulfate mass and AOD has been used to convert the sulfate mass estimates of Gao et al., (2007) into radiative properties (Schmidt et al., 2011), and is implicit in the linear scaling of ice core sulfate flux to AOD used by CU13 for eruptions of Pinatubo magnitude and smaller. We apply this same assumption hereafter (up to a threshold $M_*$, see Sec. 3.6), using a scaling factor $A$:

$$AOD_{550} = A\, M_{SO4} \quad \text{(for } M_{SO4} < M_*) \tag{2}$$

Time evolution of sulfate mass is emulated in EVA using a chemical box-model framework. Bluth et al. (1997) introduced a simple single-box model of stratospheric aerosol evolution. Changes in stratospheric sulfate aerosol are controlled by injections of $SO_2$ into the box, subsequent conversion of $SO_2$ into sulfate aerosols, and loss of sulfate aerosols to the troposphere. Describing the production and loss of sulfate mass ($M_{SO4}$, all masses in Tg S) as a function of the $SO_2$ mass ($M_{SO2}$) and characteristic timescales $\tau_{prod}$ and $\tau_{loss}$, respectively, the time tendency of $M_{SO4}$ is:



$$\frac{dM_{SO4}(t)}{dt} = \frac{M_{SO2}(t)}{\tau_{prod}} - \frac{M_{SO4}(t)}{\tau_{loss}} \tag{3}$$

This equation describing the time evolution of $M_{SO4}$ can be solved analytically, for example for a single pulse injection of $M_{SO2}$ at time $t_0$:

$$M_{SO4}(t) = M_{SO2}(t_0)\left[1 - \exp\left(\frac{-t}{\tau_{prod}}\right)\right]\exp\left(\frac{-t}{\tau_{loss}}\right) \tag{4}$$

Equations 2 and 4 can be used to emulate the observed global mean AOD evolution after the Pinatubo eruption. Using the best estimate of a total $SO_2$ injection of 9 Tg S (Guo, 2004), the parameters A, $\tau_{prod}$ and $\tau_{loss}$ can be determined based on

comparison with the CCMI global mean AOD anomaly timeseries (Fig. 1). Given the uncertainties in the CCMI AOD in the first months after the Pinatubo eruption, due to the gap in the SAGE II observations due to saturation effects (Thomason and Peter, 2006), we have based our fit on the CCMI AOD beginning in July 1992, when SAGE II retrievals of the full tropical stratosphere resumed. Therefore, the fit is not strongly constrained by the peak of global mean AOD in the CCMI reconstruction, but rather by the shape of the AOD decay. Best fit is achieved with values of $A$=0.0364, $\tau_{prod}$= 180 d, and

$\tau_{loss}$=330 d. The resultant value of A is comparable to that suggested by (Stothers, 1984) (0.0267 when converted into units of mass S rather than mass sulfate aerosol) and the stratospheric loss rate ($\tau_{loss}$=330 d) is consistent with the decay rate of AOD after Pinatubo noted by other researchers (e.g., Bluth et al., 1997). The best fit "production" rate, $\tau_{prod}$=180 d, is perhaps surprising, given its discrepancy from the observed timescale of $SO_2$ decay, which is around 30-35 d (Bluth et al., 1992; Read et al., 1993). In their single-box aerosol model, Bluth et al. (1997) noted the resulting lag between peak observed AOD

and the peak in modeled $SO_4$ mass when using an $SO_4$ production timescale equal to the observed $SO_2$ decay rate, and proposed that the rate of $SO_4$ aerosol may be limited by other chemical steps other than the destruction of $SO_2$. We contend that global mean AOD may further depend on the timescale of spatial spreading of the aerosol cloud, since the impact of aerosol contained within a horizontally contained vertical column will be diminished due to shielding effects. Whatever the mechanism responsible, $\tau_{prod}$ should be interpreted as an effective production timescale, which incorporates not only the

chemical conversion of $SO_2$ to $SO_4$, but other processes which damp the rise in AOD. An important caveat of this construction is that the peak loading of the simulated $SO_4$ mass is significantly less than what would result from a complete conversion of $SO_2$ to $SO_4$ prior to any loss. For this reason, the scaling factor $A$ is larger than what would be deduced if the peak $SO_4$ loading is assumed to be equal to the $SO_2$ injection (in Tg S).

The global mean effective radius is computed based on a simple scaling argument. For a given mass of sulfate $M_{SO4}$ which is

distributed equally among $N$ aerosols of radius $r$,

$$M_{SO4} \sim Nr^3. \tag{5}$$



If $N$ is constant, then particle radius scales as the 1/3 power of $M_{SO4}$. This relationship is similar to that used by CU13, who based $r_{eff}$ on the 1/3 power of $AOD_{550}$. This follows if $r$ is linearly related to $r_{eff}$, which is the case, for instance, if the aerosol particles are distributed log-normally by size (with a given shape parameter). In EVA, we set:

$$r_{eff} = R(M_{SO4})^{\frac{1}{3}} \qquad (6)$$

and find a scaling constant $R$ that produces best agreement in terms of the peak global mean $r_{eff}$ reached after the Pinatubo
eruption in the $r_{eff}$ timeseries of the Sato/GISS and CU13 reconstructions. Like CU13, we also set a minimum $r_{eff}$ of 0.2 µm. With a fit value of $R=0.78$, the resulting EVA $r_{eff}$ timeseries shows reasonable agreement in peak magnitude compared to the Sato/GISS and CU13 reconstructions, with peak values of between 0.5 and 0.6 µm (Fig. 1b). Basing $r_{eff}$ directly on the sulfate mass in this way does not allow the reproduction of some observed features of the $r_{eff}$ evolution apparent in the Sato/GISS $r_{eff}$ reconstruction, including the lag of its peak compared to that of $AOD_{550}$. However, given the simplicity of this
approach, and the uncertainties in $r_{eff}$ estimates retrieved from satellite sensors, the scaling methodology appears to produce satisfactory results, and even if the Sato/GISS evolution of $r_{eff}$ is more realistic, it remains to be demonstrated that this difference has any detectable influence on the climate.

### 3.3 Spatio-temporal structure

The stratosphere can be separated into regions with distinct dynamical regimes (Plumb, 1996, 2002). One important
distinction is that of the relatively undisturbed "tropical pipe"—where mean residual motion is predominantly upward—from the extratropics, where wave breaking leads to quasi-horizontal mixing, motivating the term "surf zone" (McIntyre and Palmer, 1983). The structure of stratospheric trace gas species are well reproduced by the so-called "leaky pipe" model of the stratosphere, which differentiates between the different dynamical regimes of the tropics versus extratropics (Ray et al., 2010). Following these studies, and motivated also by the clear separation of tropical vs extratropical stratospheric aerosol
maxima following the 1991 Pinatubo eruption ((Trepte et al., 1994), and Fig. 2), EVA uses a simple three-box representation of the stratosphere, separating it into three regions, equatorial, NH extratropical and SH extratropical, and describes the stratospheric aerosol distribution as the superposition of three zonally symmetric, global scale aerosol "plumes".

The aerosol properties of each plume are defined using a static characteristic spatial structure based on the 4-year average aerosol extinction from the CCMI reconstruction following the 1991 Pinatubo eruption. To reproduce the observed
meridional structure of AOD, we choose for simplicity Gaussian functions, fit as a function of the area conserving coordinate $\sin(\varphi)$, where here $\varphi$ denotes latitude. The observed evolution of zonal mean $AOD_{550}$ after Pinatubo (Fig. 2a) shows a clear separation between the three regions, with an initial growth of $AOD_{550}$ within the equatorial region, and later peaks in the NH and SH. We base the structure of the equatorial shape function on the CCMI $AOD_{550}$ averaged over the first 3 months after Pinatubo, before much transport to the extratropics occurred, and fit a Gaussian function to the central portion of the
$AOD_{550}$ (Fig. 2b). (While the magnitude of AOD550 in the first months is assumed to be highly uncertain during the first





months due to saturation effects, we necessarily assume here that the latitudinal structure of the CCMI AOD is reasonably realistic). Subtracting the Gaussian fit of the equatorial plume from the 4-year mean $AOD_{550}$ shown in Fig. 2b isolates the spatial structure of the remaining extratropical regions (Fig. 2c). We fit the observed extratropical AOD structure with Gaussian functions centered at 45° with width of 14°. At high latitudes, where the impact of our choice of the $\sin(\varphi)$

coordinate is strongest, the EVA fit shows some disagreement with the CCMI AOD, but much better agreement with the older Sato/GISS reconstruction (not shown). Given that SAGE observations are generally limited to ±60° and high latitude values are extrapolated, and that high latitude aerosol often resides at heights below 15 km which can be difficult for satellite sensors to retrieve (Ridley et al., 2014), it does not appear warranted to change the $\sin(\varphi)$ fitting procedure adopted at lower latitudes.

The three horizontal shape functions are normalized by their respective global area weighted mean. In this way, multiplication of a horizontal shape function by the sulfate mass in the region gives, for each latitude, a representation of the local vertically integrated sulfate mass density (kg/km$^2$).

The time evolution of the $AOD_{550}$ for each region is based on expanding the single box model of the stratosphere of Sec 3.2 into a three-box model. In addition to injections of $SO_2$, and loss of sulfate through cross-tropopause transport—as in the

global single box model—sulfate is transported between the three boxes, with time constants $\tau_{mix}$ and $\tau_{res}$ defining the rates of two-way mixing and poleward residual circulation, respectively. For example, the rate of change of sulfate mass in the NH region is related not just to $SO_2$ to $SO_4$ conversion and loss, but also to mixing, which is related to the difference of mass between the NH and equatorial plumes, and the transport of mass from the equatorial plume due to the residual mass circulation of the Brewer-Dobson circulation:

$$\frac{dM_{SO4}^{NH}}{dt} = \frac{M_{SO2}^{NH}}{\tau_{prod}} - \frac{M_{SO4}^{NH}}{\tau_{loss}} + \frac{\left(M_{SO4}^{EQ} - M_{SO4}^{NH}\right)}{\tau_{mix}} + \frac{M_{SO4}^{EQ}}{\tau_{res}} \tag{7}$$

Similar expressions are used for the SH and equatorial regions. In the EVA module code, the differential equations describing the time tendency of $SO_4$ within each region are computed numerically through a forward Euler method, i.e.:

$$M_{SO4}(t + 1) = M_{SO4}(t) + \Delta M_{SO4}(t). \tag{8}$$

Each eruption is treated as an instantaneous injection of $SO_2$ into one of the three boxes, with the injection region based on the latitude of the volcano, with latitudinal boundaries set to ±25 °, based on satellite-based estimates of the "edges" of the

stratospheric tropical pipe (Neu, 2003). In this way, the impact of multiple eruptions occurring with overlapping time periods of impact can be easily handled, as each eruption simply adds to pre-existing sulfur loading.

The observed $AOD_{550}$ after Pinatubo shows clear influence of the seasonal cycle of stratospheric transport, with maximum extratropical AOD found in the winter and spring seasons of each hemisphere, qualitatively consistent with the seasonal cycle of the Brewer Dobson circulation which maximizes in the winter months (Holton et al., 1995). We therefore vary the

$\tau_{mix}$ and $\tau_{res}$ parameters with calendar month m, using simple sinusoidal relationships, e.g.,



$$\tau_{\text{mix}}^{\text{NH}}(m) = \overline{\tau}_{\text{mix}}\left[1 + B\cos\left((m-1)\frac{\pi}{6}\right)\right], \qquad m \in \{1,2,\dots,12\} \tag{9}$$

where the factor $B$ describes the amplitude of the seasonal variation. Mixing and residual transport in the NH are thus strongest in January, and weakest in July, and have an annual average equal to $\overline{\tau}_{\text{mix}}$ and $\overline{\tau}_{\text{res}}$, respectively. Mixing and residual circulation in the SH are phase-shifted by 6 months, with maximum mixing in July and minimum in January.

Fitting the parameters $B$, $\tau_{\text{mix}}$, and $\tau_{\text{res}}$ was achieved by minimizing the root-mean-square residuals of the EVA $AOD_{550}$ with
the CCMI $AOD_{550}$ field for Pinatubo. Owing again to the larger uncertainties in satellite-based aerosol properties in the initial months after Pinatubo due to saturation effects, the fitting procedure ignored the initial 12 months between the latitudes between 20°S and 20°N. Best fit was achieved with values of $\tau_{mix}$ = 15 months, $\tau_{res}$ = 17 months and $B$=0.75, resulting in good agreement with CCMI $AOD_{550}$ evolution (Fig. 3). The $AOD_{550}$ evolution of EVA lacks the fine detail of the CCMI data set, but reproduces the general spatiotemporal structure, including the double peak structure in the SH
midlatitudes, and the gradual shift from strongest $AOD_{550}$ in the tropics to the extratropics with time.

In the vertical dimension, observations show that volcanic aerosol extinction values were strongly peaked in the tropics at about 22 km in the first months after the Pinatubo eruption, and thereafter spreading somewhat with height, with the peak shifting slightly downwards (Arfeuille et al., 2013). Transport of aerosol from the equatorial region to mid and high latitudes was episodic in the first year after Pinatubo, with contributions from both the lower and upper branches of the Brewer-
Dobson circulation (Trepte et al., 1993). Horizontal transport of passive dynamical tracers in the midlatitude stratosphere takes place predominantly along isentropic surfaces, due to large scale mixing processes resulting from wave activity. Aerosol transport is complicated by additional processes, including gravitational settling and anomalous vertical motions resulting from the heating of the local atmosphere by the absorption of long wave radiation by the aerosols themselves (Rogers et al., 1999). Nonetheless, the variation of peak aerosol extinction in the four-year post-Pinatubo mean follows
isentropic surfaces (derived from ERA interim reanalysis data, (Dee et al., 2011)) reasonably closely in the mid latitudes of the summer hemisphere, with the 430 K potential temperature surface corresponding well with the vertical peak of mean EXT (Fig. 4a). The correspondence between the aerosol loading and isentropic (potential temperature) surfaces breaks down in the high latitudes during winter, where diabatically driven vertical motion within the polar vortex leads to downwelling of air parcels with respect to potential temperature (Manney et al., 1994; Tegtmeier et al., 2008).

To reproduce the variation of the vertical peak of extinction with latitude in the extra-tropics, we define a vertical "center-line" based on climatological potential temperature. While potential temperature does not perfectly follow the observed aerosol peak at high latitude, linking the vertical distribution of the aerosol to a temperature (rather than mass or geopotential) based vertical coordinate, it may be more suitable for application in much warmer or colder climates. Specifically, we define the center-line at mid and high latitudes ($\varphi > 45°$) from the summer 430 K potential-temperature
surface for each hemisphere (July for NH, January for the SH), and the annual mean 430 K potential-temperature surface for $-45° < \varphi < 45°$. This empirically defined center-line shows reasonable agreement with the observed vertical peak in aerosol



extinction in the extra-tropics (Fig. 4a). In the tropics, the vertical position of the plume is given a vertical offset from the center-line.

The vertical shape of extinction in the equatorial region is based on a Gaussian fit of the 4-year Pinatubo mean CCMI extinctions, with a vertical width of 2.25 km and a vertical offset of 2.75 km (Fig. 4b). In the extratropics, a Gaussian fit is

produced, centered on the defined center-line, with a width of 2.825 km (Fig. 4c). The vertical shape functions, defined internally on a 1-km vertical grid, are normalized by their vertical sum. Therefore, multiplication of the AOD at each latitude by the normalized vertical shape function produces a profile of aerosol extinction (km$^{-1}$).

The resulting EVA aerosol extinction structure shows good agreement with the four-year mean CCMI Pinatubo observations (Fig. 5). The use of Gaussian vertical and horizontal shape functions results in fairly good reproduction of the strong

extinction gradient at the tropopause.

The spatiotemporal structure of aerosol effective radius is based on the simulated sulfate mass timeseries for each latitude (i.e., a function of the three-box model and the horizontal shape functions). Effective radius is computed for each latitude and time from Eq. 6, and assumed to be constant with height through the stratosphere. The resulting $r_{eff}$ field shows reasonable agreement with observation-based estimates (Fig. 6), however, appears to produce peak values which are

somewhat too large, and peak too early compared to the Sato/GISS reconstruction, as apparent also in the global mean (Fig. 1).

## 3.4 Wavelength dependent optical properties

The wavelength dependent optical properties EXT, SSA and ASY are computable via Mie scattering theory given knowledge of the extinction at a specific wavelength, and the effective radius of an assumed log-normal size distribution.

EVA utilizes look-up tables, computed from Mie theory, assuming a single-mode log-normal size distribution with width parameter $\sigma$=1.2, which give wavelength dependent EXT scaling ratios (with respect to EXT at 550 nm), ASY and SSA for varying effective radii, ranging from 0.2 to 1.3 μm, and for 29 wavelengths ranging from 0.2 to 100 μm. Therefore, given the EXT$_{550}$ and $r_{eff}$ at any point in space, EXT($\lambda$), SSA($\lambda$) and ASY($\lambda$) are calculated from the lookup tables through bilinear interpolation. The use of $\sigma$ =1.2 is roughly consistent with that deduced from observations of the Pinatubo aerosol cloud,

with (Stenchikov et al., 1998) using $\sigma$=1.25 and the CCMI reconstruction using $\sigma$=1.2 in gap-filled regions when the SAGE measurements cannot be used to estimate sigma directly (Arfeuille et al., 2013).

Resulting zonal mean AOD at four sample wavelengths are shown in Fig. 7, and compared to the corresponding CCMI values. In general, EVA reproduces the decrease of AOD with increasing wavelength from the visible to the near infrared. SSA and ASY at four wavelengths, at 20 km, are shown in Figs. 8 and 9, and again, reproduce reasonably well the variation

of magnitude with changing wavelength. Some differences in the spatial structure of SSA are apparent between the EVA and CCMI reconstructions, although it remains to be shown how important such structure is in the overall climate impact of the volcanic forcing.



### 3.5 Background aerosol forcing

Satellite observations of aerosol extinction in the years following the 1991 Pinatubo eruption show that the extinction approaches a non-zero minimum value. The assumption that radiative forcing by stratospheric aerosol would decay to zero in the 2000-2010 decade has been shown to lead to non-negligible biases in climate model simulations (Solomon et al., 2011).

The background stratospheric aerosol layer is thought to be a result of a combination of factors, including the episodic but ubiquitous influence of relatively minor volcanic eruptions with small stratospheric injections, and the slow and steady influx of aerosols and precursors from the troposphere into the stratosphere (Kremser et al., 2016).

A non-zero background stratospheric aerosol forcing is parameterized within EVA by specifying a constant $SO_2$ injection value, chosen to produce a best fit between the CCMI and EVA global mean AOD timeseries for the year 2000, where the

observed global AOD reached a minimum (Fig. 10). This simple procedure results in a background $SO_2$ injection estimate of 0.2 Tg y$^{-1}$. For comparison, using the coupled aerosol Solar-Climate Ozone Links chemistry climate model (SOCOL-AER) and CCMI retrievals, (Sheng et al., 2015) have inferred a total net sulfur mass flux into the stratosphere of 0.19 Tg y$^{-1}$.

The structure of the observed aerosol extinction in the meridional height plane during the 2000 minimum is shown in Fig. 11a. Maximum aerosol extinctions in the CCMI data set are in the high-latitude lowermost stratosphere. To best reproduce

this structure, the background $SO_2$ injections are split evenly between the two extratropical plumes. The resulting structure of EVA background AOD forcing does not reproduce the lower stratosphere forcing due to the static shape functions of the EVA construction based on the Pinatubo aerosol evolution. Nonetheless, this strategy of background AOD in EVA does somewhat reproduce the meridional structure of the background AOD (Fig. 11c,d).

### 3.5 Hemispheric asymmetry

Some tropical eruptions, like Pinatubo (1991), lead to relatively even partitioning of between the NH and SH, but others, like Agung (1963) and El Chichón (1982), are characterized by substantial hemispheric asymmetry in stratospheric aerosol loading. Asymmetries in the radiative forcing, if large enough, may have a detectable climate impact, for instance through their influence on the latitude of the inter-tropical convergence zone and subsequent changes in tropical monsoon patterns (Haywood et al., 2013; Oman et al., 2006).

Some degree of hemispheric asymmetry may be expected due to the timing of an eruption with respect to the seasonal cycle of stratospheric transport. One expects tropical eruptions occurring in NH winter, when transport to the NH extratropics is strongest, to produce higher NH aerosol loadings than those in summer. This expectation is reproduced by aerosol general circulation models (Toohey et al., 2011). The seasonal variation of stratospheric transport parameterized within EVA leads to a small degree of hemispheric asymmetry in the resulting AOD patterns depending on season of eruption, with highest

asymmetry produced for tropical eruptions in August ($AOD_{NH}/AOD_{SH}=1.18$) and February ($AOD_{NH}/AOD_{SH}=0.847$).

The parameterized seasonal stratospheric transport of EVA qualitatively reproduces the observed SH bias of the AOD from the February 18, 1963 Agung eruption (Fig. 12). However, quantitative estimates of the SH bias to the Agung AOD are



stronger than that produced by EVA. For instance, based on ground-based optical measurements, (Stothers, 2001) estimated that the SH aerosol loading after Agung was 8 times larger than that of the NH. Furthermore, the hemispheric asymmetry produced by the seasonal stratospheric transport of EVA for El Chichón, for an April 4 eruption date, is much different than the observed strong NH asymmetry (Fig. 12).

For these reasons, an anomalous asymmetry factor is included in EVA to allow one to impose an observed asymmetry from data. Hemispheric asymmetry can be specified in the input file, based on direct observational estimates or estimated from the ratio of Greenland to Antarctic ice core sulfate records. When this ratio differs from the asymmetry produced by the seasonal mixing parameterization of EVA, a correction can be applied, which attenuates the rate of mixing and transport to one hemisphere for a period of 18 months after the eruption. This correction ensures that the AOD in the mid and high latitudes

shows a hemispheric ratio equal to that of the input value, while retaining the nominal seasonality of stratospheric dynamics. Hemispheric asymmetry corrections are applied here to the EVA reproductions of the Agung, Fuego and El Chichón eruptions (Table 1). In order to enhance comparability with the CCMI data set, correction factors for Agung and Fuego are based on the AOD reconstruction of (Stothers, 2001), and defined as the ratio of the 2-year post-eruption average of hemispheric AOD. This method produces values of 0.19 for Agung, and 1.0 for Fuego. For El Chichón, we calculate a

correction factor directly from the CCMI $AOD_{550}$, again as the ratio of the 2-year AOD average for each hemisphere, resulting in a value of 1.5. Applying hemispheric asymmetry corrections in the reconstruction of $AOD_{550}$ for Agung and El Chichón produces better agreement with the CCMI estimates (Fig. 12). It should be noted here, that the CCMI representations for these eruptions are highly uncertain. For Agung, the CCMI data is based on 2D model results scaled by sparse ground-based measurements, while for El Chichón, it is based on an amalgam of very sparse ground and airplane-

based lidar and satellite observations at only the high latitudes (Thomason and Peter, 2006).

### 3.6 Nonlinear scaling for large eruptions

While a simple linear relationship between sulfate mass and AOD (Sec 3.2) seems not inconsistent with available observations, its applicability to eruptions larger than Pinatubo is quite uncertain. Modeling studies imply that for large eruptions, the maximum AOD produced is not a linear function of the injected sulfur (Timmreck et al., 2010). CU13 argued

that above some threshold, AOD should scale as the 2/3 power of sulfate aerosol mass rather than linearly. Such a relationship is consistent with the results of an aerosol-general circulation model simulating a large range of tropical volcanic eruptions (Metzner et al., 2012). We sketch a simple explanation for this relationship as follows: for a total mass of sulfur (*M*) distributed among *N* particles with uniform radius *r*, the mass of each particle, *M/N*, is proportional to the volume of each particle (*V*) and therefore $r^3$:

$$\frac{M}{N} \sim V \sim r^3 \tag{10}$$

The AOD is proportional to the total cross sectional area of the particles:



$$AOD \sim Nr^2 \tag{11}$$

Combining these equations, AOD can be written as a function of N and M:

$$AOD \sim N^{1/3}M^{2/3} \tag{12}$$

When new particles are formed by a sulfur injection in proportion to the mass of injected sulfur, $N$ is proportional to $M$ and the AOD scales linearly with $M$. On the other hand, if injected sulfur mass condenses onto pre-existing particles, $N$ remains constant, and AOD scales with the 2/3 power of M. These two regimes—complete new particle formation from the injected $SO_2$ or complete condensation onto pre-existing particles —obviously represent the two extremes of possible sulfate aerosol evolution, and it seems likely that both processes take place in differing degrees. Although it might be more physical to parameterize $N$ as, for instance, the logarithm of $M$, in EVA we adopt a threshold-based implementation similar to what was done by CU13. We retain this approach for very large eruptions, and scale sulfate to AOD based on two parameters, $A_1$ and $A_2$, such that:

$$AOD = \begin{cases} A_1 M_{SO4}, & M_{SO4} < M_* \\ A_2 M_{SO4}{}^{2/3}, & M_{SO4} \geq M_* \end{cases} \tag{13}$$

Where $M_*$ is defined so as to make the relationship continuous, i.e.:

$$M_* = \left(\frac{A_2}{A_1}\right)^3 \tag{14}$$

In EVA, the $A_1$ term is based on the peak global mean AOD from the CCMI data set and the best estimate of 9 Tg S injection from Pinatubo (Sec 3.2). We choose to define $A_2$ so as to best reproduce the aerosol forcing of CU13 for large eruptions, such as Tambora and Samalas.

Figure 13 shows the relationship between peak global mean AOD and estimated injected $SO_2$ for Pinatubo, Tambora and Samalas. The linear scaling of CU13, which was based on the AOD estimates of Sato and ice core-based estimates of $SO_2$ injection, leads to a rather steep curve, which, when extrapolated to the estimated sulfur injection of Tambora, would have produced an estimate of global mean AOD of about 0.7. A much smaller AOD for Tambora (i.e., 0.45) and Samalas was produced by CU13 by incorporating the 2/3 power law, which they applied to eruptions larger than Pinatubo, producing the AOD values shown in Fig. 13.

The linear scaling used in EVA is significantly less steep than that of CU13, a result of the lower peak global mean AOD estimate for Pinatubo from CCMI compared to Sato/GISS, and the larger estimate of $SO_2$ injection from satellite sensors compared to the ice core-derived estimate of CU13. Extrapolation of the linear scaling of EVA to larger injection magnitudes reproduces rather well the CU13 estimates of AOD and $r_{\text{eff}}$ for Tambora. A 2/3 power law relationship is



nonetheless implemented in EVA, which applies then to eruptions greater in magnitude than Tambora, and therefore reduces the impact of the Samalas eruption compared to using the linear relationship.

## 4. Sample Results

### 4.1 Modern era

AOD timeseries produced by EVA using the eruption history of Table 1 are shown in Fig. 14, and compared to the CCMI data set. The magnitude of global mean AOD of the major eruptions is well reproduced by EVA, with slightly larger AOD produced for Pinatubo (as discussed in Sec 3.2) and slightly lower AOD for Agung compared to the CCMI data, which relies on ground-based optical data to scale the AER model results for Agung, Fuego and El Chichón. A number of relatively minor eruptions before and after the El Chichón eruption (Bluth et al., 1997)–not accounted for in the eruption history used

here-- likely account for the complex temporal and spatial AOD evolution seen in the CCMI data base in the years 1980-1990.

Including the minor eruptions of 2000-2014 in the EVA input file improves the comparison over these years. This time period is examined in more detail in Fig. 15. The EVA global mean AOD tracks the variability shown in the CCMI database (based in this time period on CALIPSO satellite data) rather well. The EVA global mean AOD response to the high latitude

eruptions of Kasatochi (2008) and Sarychev (2009) exceeds that seen in the CCMI data, and appears to persist longer. This is likely a result of the use of a single aerosol loss rate for all eruptions in EVA, which would not take into account reduced lifetimes due to injection into the lowermost extratropical stratosphere. This highlights a limitation of the EVA approach, and suggests accurate reproduction of small magnitude eruptions may require additional considerations. Nonetheless, we note that the agreement between EVA and CCMI is decent at ~60°N, the northernmost limit of the CALIPSO measurements

underlying the CCMI data, implying that the stronger apparent disagreement at polar latitudes may be due partly to the gap-filling procedure used in the construction of the CCMI dataset. The agreement between EVA and CCMI over this time period is also largely dependent on the accuracy of the stratospheric injection estimates used. The estimates used here, based on MIPAS satellite $SO_2$ measurements, carry an uncertainty on the order of 20% (Höpfner et al., 2015), with additional (unquantified) uncertainty arising from the separation of the stratospheric component of the total atmospheric injection

(Brühl et al., 2015). If desired, agreement between the EVA-based results with the satellite records over the 200-2014 period could be improved by careful adjustment of the input sulfur injections.

### 4.2 Tambora

Global mean $AOD_{550}$ and $r_{eff}$ produced by EVA for the 1815 eruption of Tambora are compared to the reconstruction of CU13 in Figure 16. Using the linear scaling, based on the estimated peak AOD and sulfur injection of Pinatubo, the peak

AOD of an estimated 55 Tg $SO_2$ injection by Tambora leads to a smaller global mean AOD in the EVA reconstruction



compared to CU13. The EVA AOD peaks at approximately 0.35, while the CU13 reconstruction estimates a global mean AOD peak of ~0.4.

It should be noted, that with the Pinatubo-based linear relationship used in EVA, if a nonlinear, 2/3 power law relationship were implemented with the same threshold used by CU13, i.e., applying to eruptions just greater in magnitude than Pinatubo,

then the resulting AOD for Tambora would be significantly smaller than the present EVA estimate.

The aerosol effective radius produced by EVA for Tambora is similar, but slightly larger than the CU13 estimate. This difference comes about because in EVA, $r_{eff}$ is scaled according to the $SO_4$ mass, while in CU13, the AOD is used. Since in CU13, the AOD for Tambora is related to the 2/3 power of mass, the muting of the AOD affects also the $r_{eff}$, while in EVA, $r_{eff}$ grows linearly with $SO_4$ mass.

The zonal mean $AOD_{550}$ for Tambora from CU13 and from this work are shown in Fig. 17. While the magnitude of peak AOD and its temporal decay are similar in the two reconstructions, the EVA reconstruction provides a more realistic latitudinal distribution of AOD compared to the 4-band structure of CU13. The slight SH bias of the CU13 Tambora $AOD_{550}$, inferred by CU13 from ice core records, is produced by EVA as a result of the parameterized seasonal transport (Sec. 3.3), with no additional hemispheric correction.

**5. Conclusions and outlook**

EVA has a number of strengths, drawing on the advantageous aspects of previous volcanic aerosol reconstructions. Like the Stenchikov and CCMI data sets, EVA provides full field optical properties (EXT, SSA, ASY) as a function of wavelength, height and latitude, allowing consistent implementation within different climate models. Like the Amman and Gao reconstructions, EVA provides a consistent treatment of all eruptions, avoiding gaps and discontinuities that are unavoidable

in reconstructions based only on observations. EVA produces good agreement with satellite-based observations of the Pinatubo eruption, providing confidence in its ability to produce reasonably realistic forcing structure. At the same time, the construction of EVA allows for great flexibility. Different EVA reconstructions can be constructed using different histories of stratospheric sulfur injections from observations (e.g., Brühl et al., 2015; Höpfner et al., 2015; Neely and Schmidt, 2016) or from ice cores (e.g., Gao et al., 2008), thereby translating uncertainties in the volcanic emission estimates into aerosol

forcing parameters. Adjusting EVA reconstructions to be consistent with updated satellite retrievals or with model results is achievable through modification of the parameter settings. This flexibility also makes EVA well suited for idealized studies: for instance, the impact of different uncertainties in volcanic aerosol properties could be tested through producing an ensemble of EVA forcing sets with an ensemble of parameter settings. The simplicity of EVA also ensures that it is fast, and can provide forcing reconstruction instantaneously for any given past or future eruption.

By design, EVA is simple, and cannot reproduce all the features of aerosol forcing which are seen in observation-based reconstructions or aerosol-general circulation model results. The three-box representation of stratospheric transport neglects the impact of the polar vortex, which creates a mixing barrier in the winter hemisphere and likely enhances aerosol loss in a





seasonal manner. EVA also does not presently consider the height of stratospheric sulfur injection, which likely plays an important role in the timescale of cross-tropopause transport (Bluth et al., 1997). Similarly, EVA does not account for vertical variations in stratospheric dynamics, specifically the shallow and deep branches of the BDC, which have different strengths and seasonal variability.

Instead of attempting to perfectly reproduce observed aerosol properties, EVA makes it possible to pose the scientific question as to what aspects of the volcanic aerosol can produce a detectable climate response, thereby providing a means for deepening our understanding of the interaction of the stratospheric aerosol and climate. Future updates to EVA are planned, but in keeping with its original motivation, no attempt will be made to represent all aspects of the observations, only those

that can be demonstrated to have a detectable influence on the climate response. Updates will be motivated by new and updated observations and, to the extent they can reliably constrain remaining uncertainties, information from more complex aerosol models. In the meantime, in its present form, EVA could be used for a variety of purposes, including producing forcing for paleo-modelling simulations, for decadal prediction simulations in the case of a major eruption (Timmreck et al., 2016), for filling gaps in satellite-based reconstructions, or simply to assess what aspects of the stratospheric aerosol forcing

lead to a detectable climate response.

**Code Availability**

EVA version 1.0 code, a user's manual, sample input data files and driver scripts, are included as supplementary material.

**Acknowledgements**

The authors thank the many researchers who have collected and processed observations of volcanic aerosol, providing the
basis of understanding upon which this work is based. The authors also thank Alan Robock and Stephanie Fiedler for constructive comments. M. Toohey acknowledges support by the Deutsche Forschungsgemeinschaft (DFG) in the framework of the priority programme "Antarctic Research with comparative investigations in Arctic ice areas" by grant TO 967/1-1 . C. Timmreck acknowledges support from the German Federal Ministry of Education (BMBF), research program "MiKlip" (FKZ: 01LP1517B) and the European Project 603557-STRATOCLIM under program FP7-ENV.2013.6.1-2.
Computations were carried out at the German Climate Computing Centre (DKRZ). Primary data and scripts used in the analysis and other supplementary information that may be useful in reproducing the author's work are archived by the Max Planck Institute for Meteorology and can be obtained by contacting publications@mpimet.mpg.de.



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



Tables

**Table 1: Stratospheric sulfur injection estimates used in this work**

| Eruption | Year | Month | Latitude (°N) | Sulfur injection (Tg S) | Hemispheric asymmetry ratio[a] | Sulfur injection source |
|---|---|---|---|---|---|---|
| Tambora | 1815 | 4 | -8.2 | 27.5 | - | Ice core sulfate flux (Gao et al., 2008) |
| Agung | 1963 | 3 | -8.3 | 5.22 | 0.19[b] | Ice core sulfate flux (Gao et al., 2008) |
| Fuego | 1974 | 10 | 14.5 | 1.18 | 1.0[b] | Ice core sulfate flux (Gao et al., 2008) |
| El Chichón | 1982 | 4 | 17.2 | 3.5 | 1.5[c] | Ice core sulfate flux (Gao et al., 2008) |
| Pinatubo | 1991 | 6 | 15.1 | 9 | - | Satellite retrievals (Guo, 2004) |
| Manam | 2005 | 1 | -4.1 | 0.08 | - | Satellite retrievals (Brühl et al., 2015) |
| Soufriere Hills | 2006 | 5 | 16.7 | 0.07 | - | Satellite retrievals (Brühl et al., 2015) |
| Rabaul | 2006 | 10 | -4.3 | 0.08 | - | Satellite retrievals (Brühl et al., 2015) |
| Kasatochi | 2008 | 8 | 52.2 | 0.19 | - | Satellite retrievals (Brühl et al., 2015) |
| Sarychev | 2009 | 6 | 48.1 | 0.28 | - | Satellite retrievals (Brühl et al., 2015) |
| Merapi | 2010 | 11 | -7.5 | 0.05 | - | Satellite retrievals (Brühl et al., 2015) |
| Nabro | 2011 | 6 | 13.4 | 0.184 | - | Satellite retrievals (Brühl et al., 2015) |

[a] Hemispheric asymmetry ratio defined as the estimated ratio of NH/SH AOD or stratospheric aerosol loading.

5    [b] Taken as the ratio of $AOD_{NH}/AOD_{SH}$, for the first two years after each eruption from the reconstruction of (Stothers, 2001).

[c] Taken as the ratio of $AOD_{NH}/AOD_{SH}$, for the first two years after the eruption from the CCMI reconstruction (Arfeuille et al., 2013; Thomason and Peter, 2006).





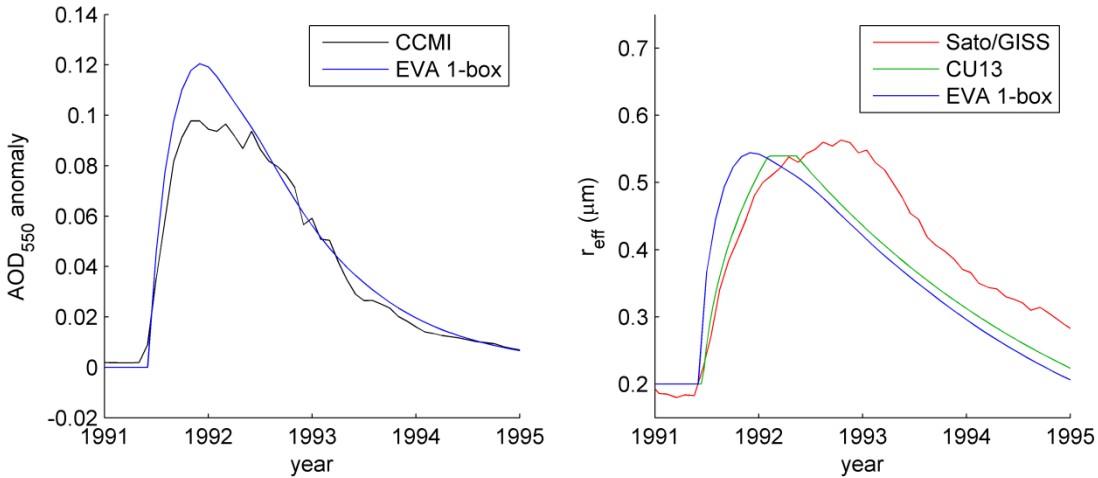

**Figure 1: (left) Global mean AOD$_{550}$ anomaly timeseries from CCMI** (Arfeuille et al., 2013) **following the Pinatubo eruption, and the reproduction via the EVA, single box model approach (see text). (right) Global mean aerosol effective radius timeseries from the reconstructions of Sato/GISS** (Sato et al., 1993)**, CU13** (Crowley and Unterman, 2013) **and the EVA single box-model approach.**





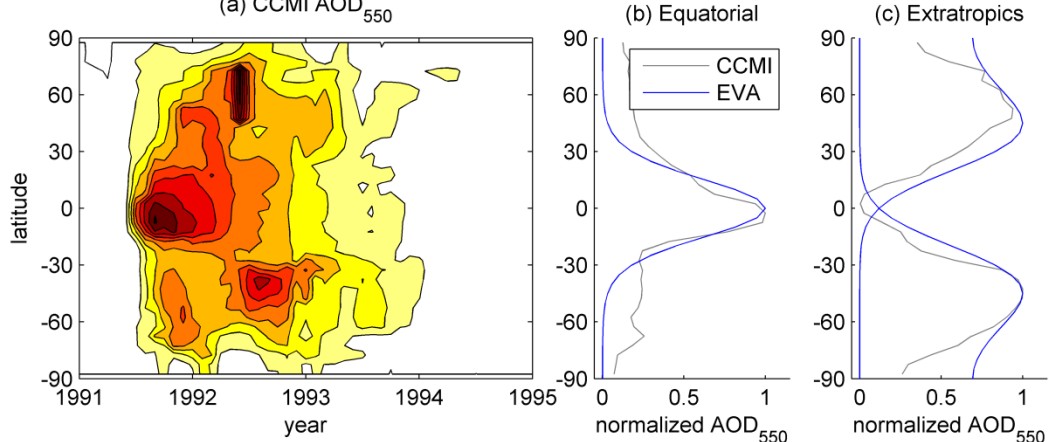

**Figure 2: Definition of latitudinal shape functions. (a) The zonal mean CCMI AOD anomaly at 550 nm as a function of latitude and time for four years. (b) CCMI AOD anomaly averages over the 2 months (gray) after the Pinatubo eruption, normalized by its maximum value. A Gaussian fit to the equatorial portion of the 2-month average (blue) is used to define the latitudinal shape function for the equatorial plume of EVA. (c) The residual of the CCMI 4 year mean AOD minus the EVA equatorial shape function (gray) is used to define the extra-tropical EVA shape functions (blue).**





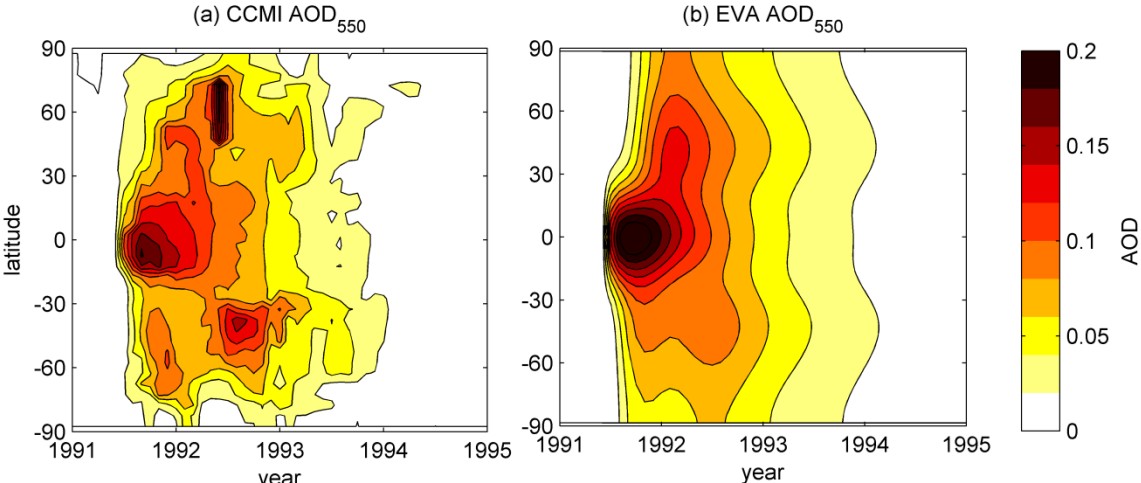

**Figure 3: Aerosol optical depth ($\lambda$=550 nm) evolution after Pinatubo from (left) the CCMI database and (right) emulated by EVA.**



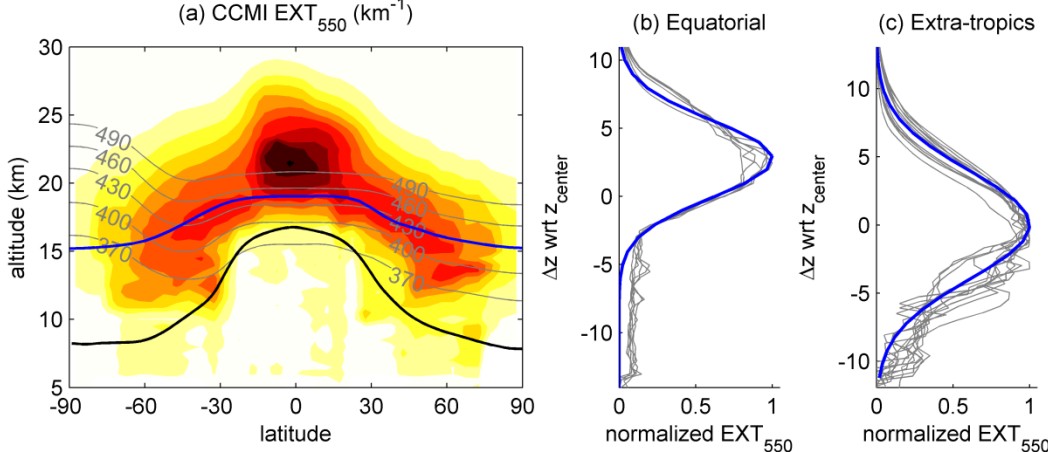

**Figure 4: Definition of EVA vertical shape functions. (a) CCMI aerosol extinction averaged over the first 4 years after the Pinatubo eruption, as a function of latitude and height. The climatological tropopause is shown in black, and climatological July potential temperature (K) surfaces shown in gray, for values as labeled. The EVA vertical centerline ($z_{center}$), as defined in text, is shown in blue. (b) Normalized CCMI/SAGE_4λ 4-year post-Pinatubo average extinction profiles as a function of altitude with respect to the vertical centerline for the tropical (-15 to 15°N) latitudes (gray), and the Gaussian fit defining the EVA vertical shape function for the equatorial plume (blue). (c) Same as (b) for extratropical (30-60°) profiles.**





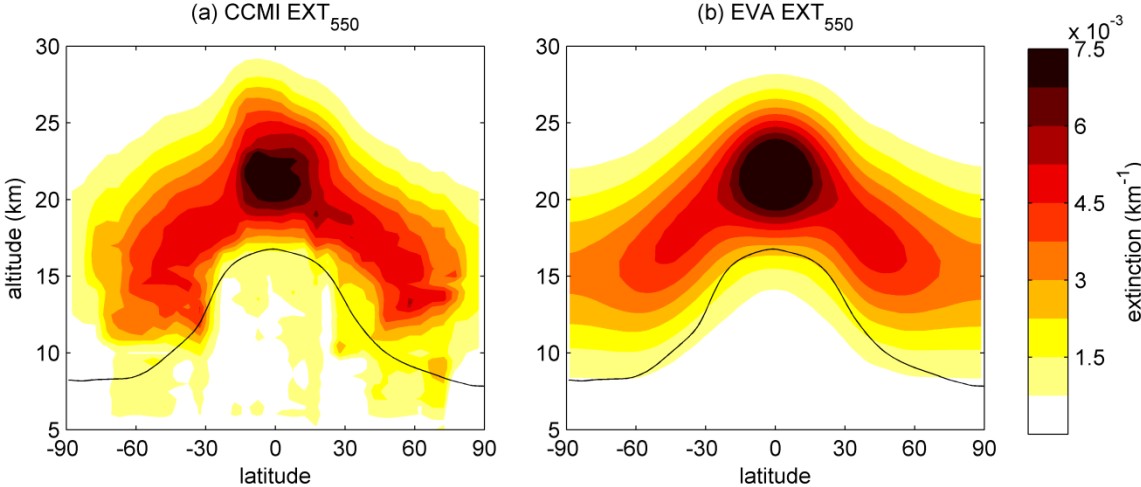

**Figure 5: Aerosol extinction (λ=550 nm) profiles for the 4-year post-Pinatubo average from (left) the CCMI database and (right) emulated by EVA.**





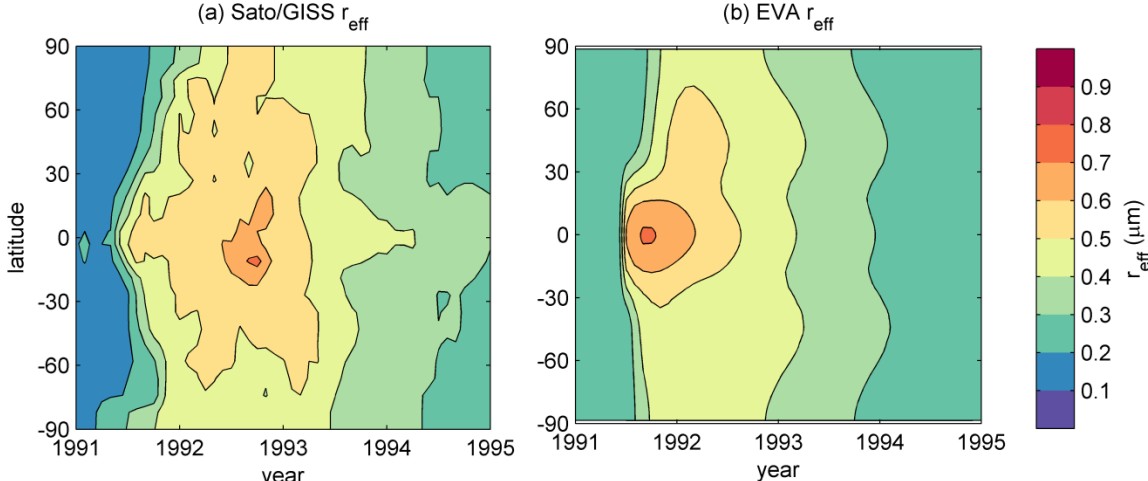

Figure 6: Zonal mean effective radius after Pinatubo from the (a) Sato/GISS reconstruction and (b) EVA.




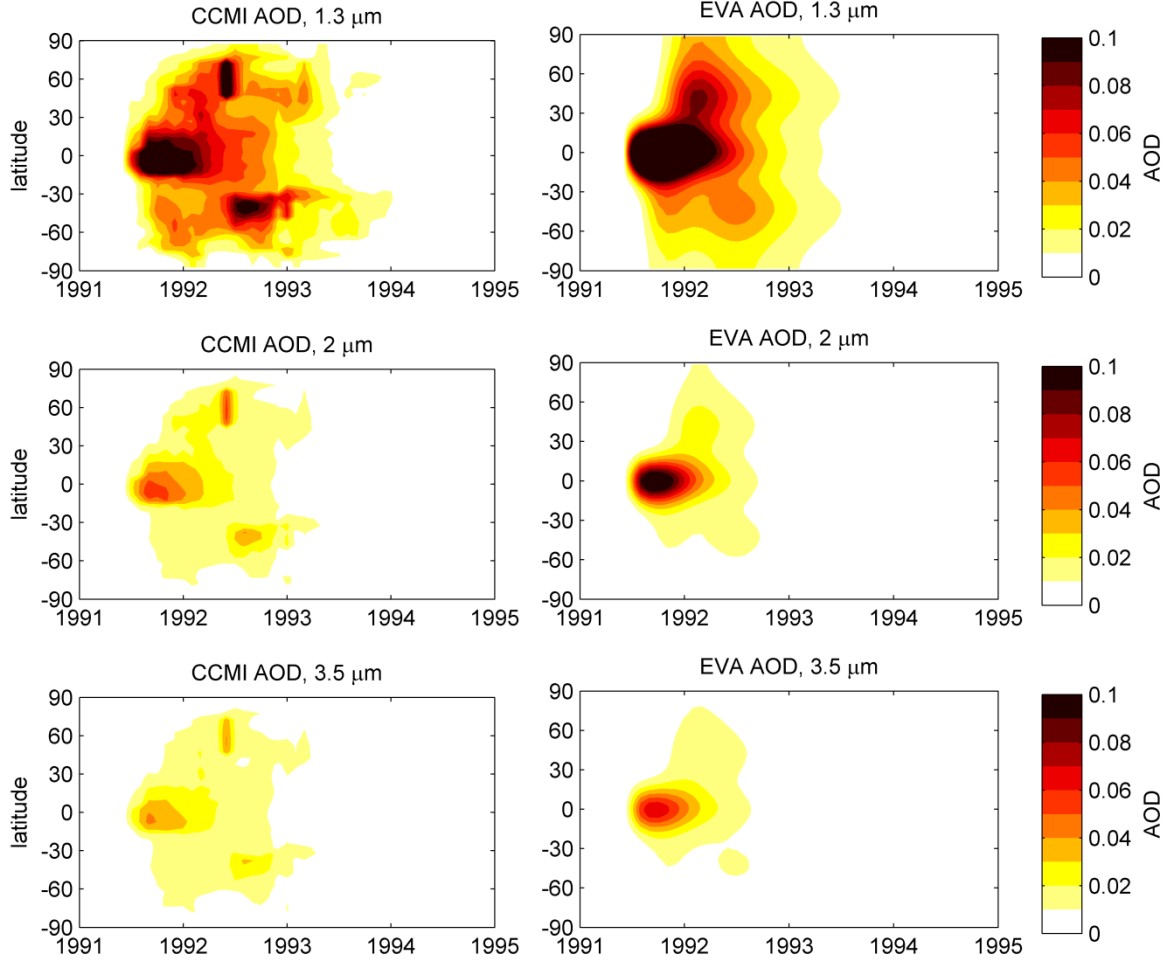

**Figure 7: Aerosol optical depth (AOD) after the Pinatubo eruption for selected wavelengths from (left) CCMI and (right) EVA.**







**Figure 8: Single scattering albedo (SSA) after the Pinatubo eruption at 20 km for selected wavelengths from (left) CCMI and (right) EVA. Note different colorscale for bottom panels.**

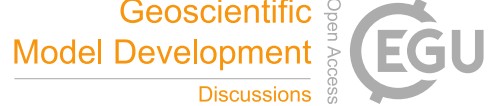







**Figure 9: Scattering asymmtery factor (ASY) after the Pinatubo eruption at 20 km for selected wavelengths from (left) CCMI and (right) EVA.**

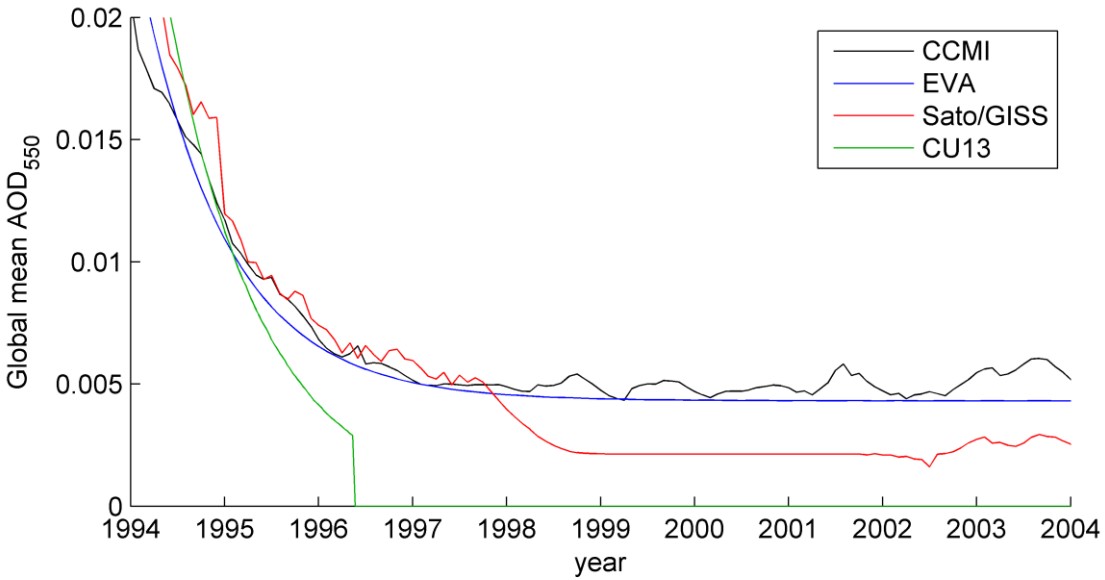

5  **Figure 10: Global mean AOD for the 1994-2004 period from the EVA single-box approach and the CCMI, Sato/GISS and CU13 reconstructions.**





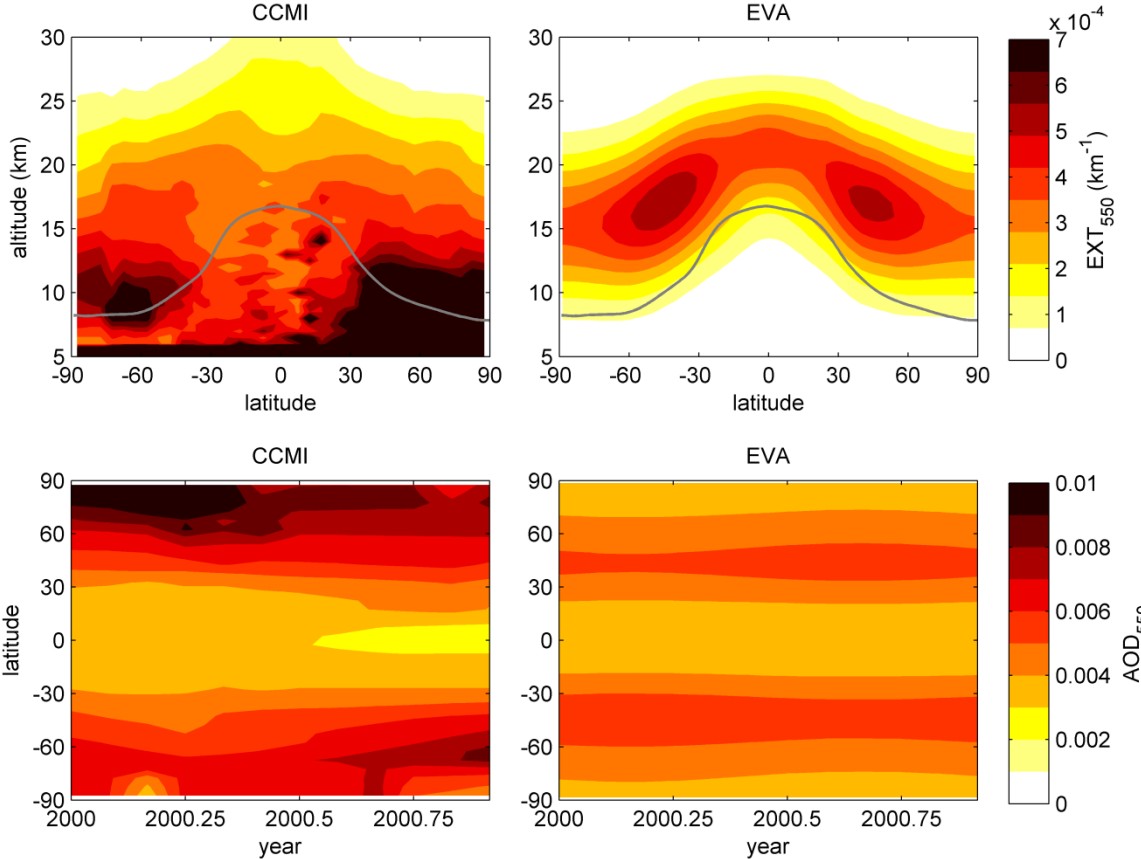

**Figure 11: (top) Aeorsol extinction at 550 nm averaged over the year 2000 from (left) the CCMI observation-based reconstruction and (right) EVA. (bottom) AOD at 550 nm through the year 2000 from (left) CCMI and (right) EVA.**





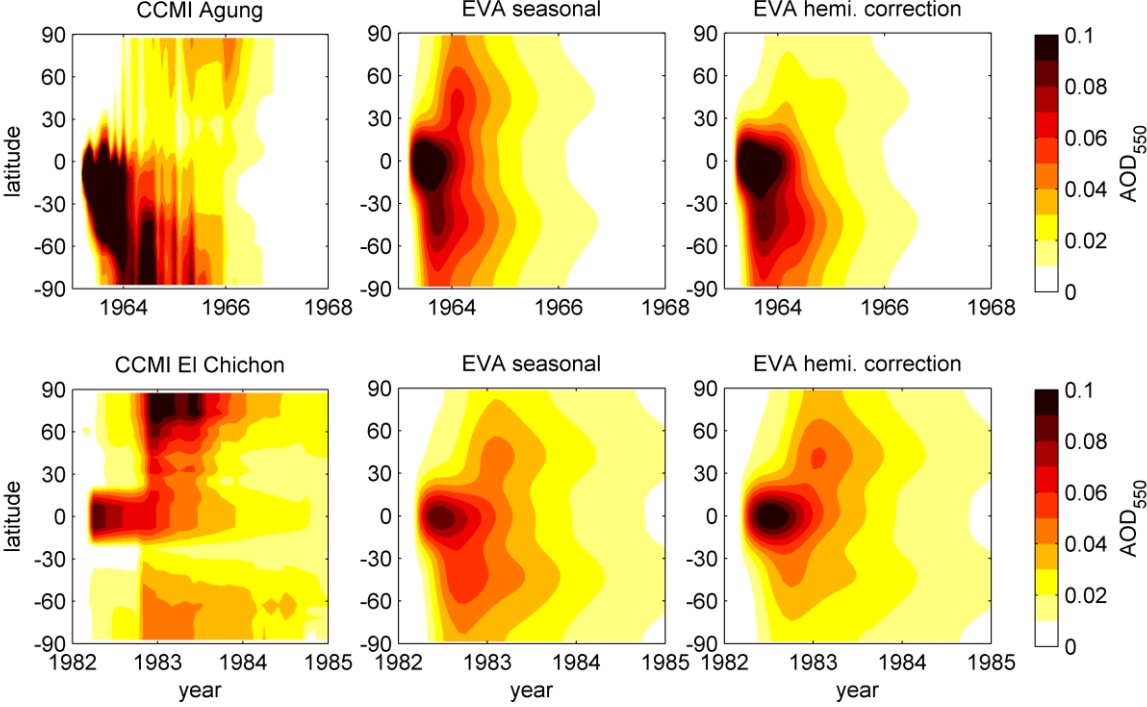

**Figure 12: Illustration of hemispheric asymmtery correction in EVA. (left) CCMI representations of the zonal mean AOD for (top) Agung and (bottom) El Chichón. EVA emulations (middle) without and (right) with hemispheric asymmtery correction, as described in the text.**





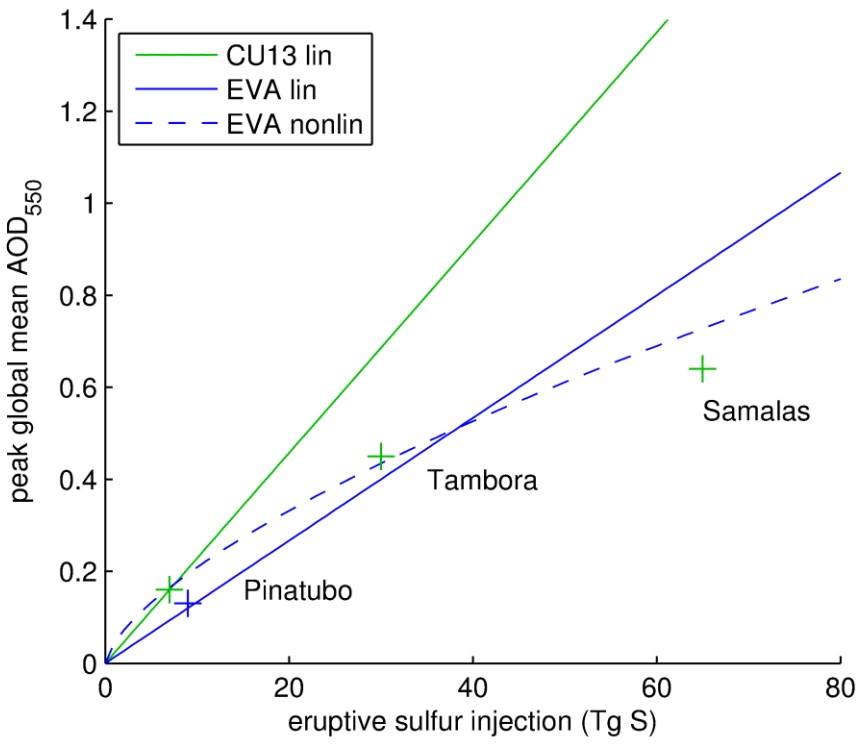

**Figure 13: Relationships between eruptive stratospheric sulfur injections and peak global mean AOD$_{550}$ from the CU13 reconstruction and EVA. Green crosses show the peak AOD values and injection estimates values from the CU13 reconstruction for Pinatubo, Tambora and Samalas. The green line shows the linear relationship which would be deduced from the Pinatubo data used in CU13, extrapolated to all injection magnitudes. The blue cross shows the peak AOD and sulfur injection estimates from satellite sensors used in EVA to construct a linear relationship, shown by the blue solid line. The blue dashed line shows the 2/3 power-law relationship used in EVA for eruptions larger in magnitude than Tambora.**




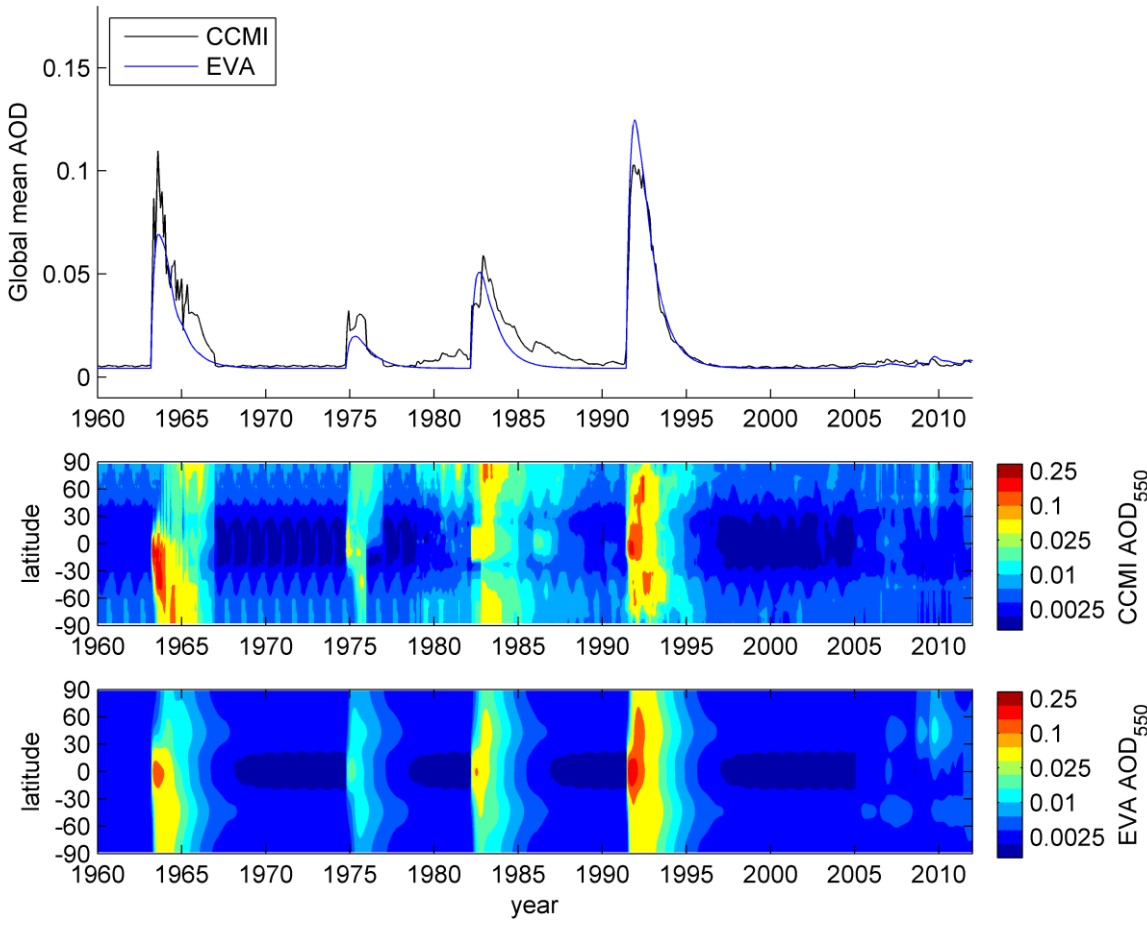

**Figure 14: Aerosol optical depth over the period 1960-2015 from the CCMI data set and from EVA, using the volcanic eruption history of Table1. (top) Global mean AOD at 550 nm from the two reconstructions. (bottom two panels) Zonal mean AOD at 550 nm as a function of latitude and time, with log color-scale.**





**Figure 15:** **Aerosol optical depth over the priod 2004-2012 from the CCMI data set and from EVA, using the volcanic eruption history of Table1. (top) Global mean AOD at 550 nm from the two reconstructions. (bottom two panels) Zonal mean AOD at 550 nm as a function of latitude and time.**





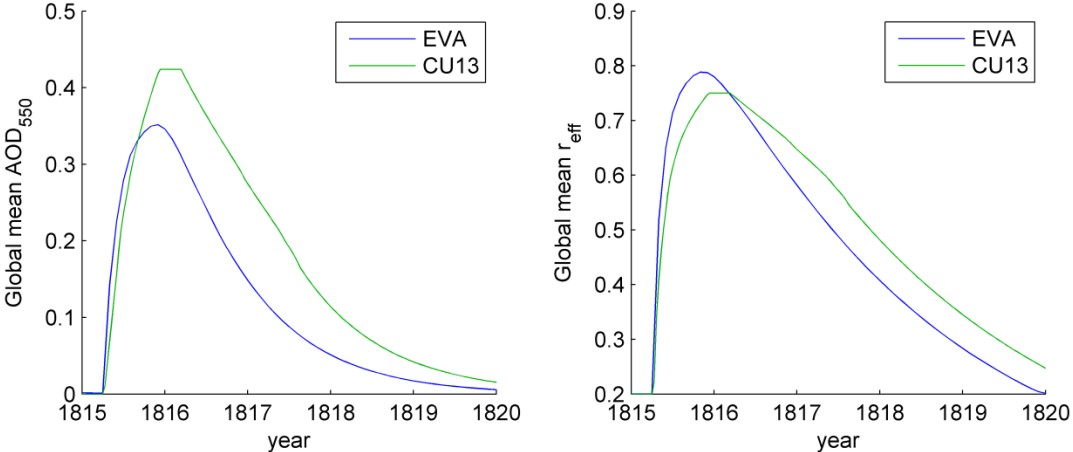

**Figure 16: Global mean AOD at 550 nm and aerosol effective radius ($r_{\text{eff}}$) for the 1815 Tambora eruption, from the CU13 reconstruction and EVA.**



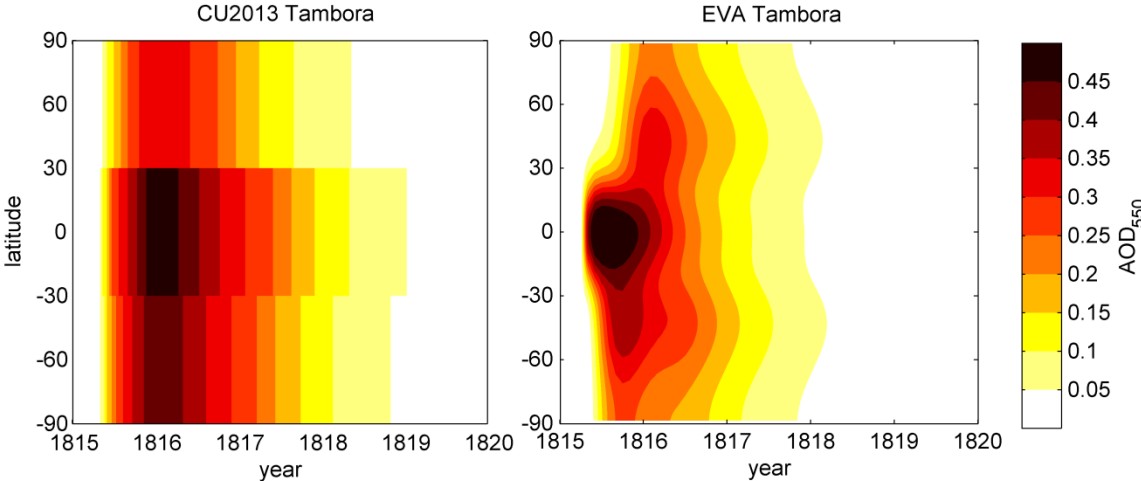

Figure 17: Zonal mean AOD(550) for the Tambora eruption from (left) CU13 and (right) EVA.

