# Peer review of "Easy Volcanic Aerosol (EVA v1.0): An idealized forcing generator for climate simulations"

_Geoscientific Model Development, 2016_

## Referee Comment (RC1) · Anonymous Referee #1 · 16 Jun 2016

Review of the manuscript

Easy Volcanic Aerosol (EVA v1.0): An idealized forcing generator for climate simulations

by M. Toohey et al.

submitted for publication in Geoscientific Model Development

**General**

The manuscript describes an algorithm for calculating temporal, latitudinal, vertical and wavelength dependent distribution of sulfate aerosols for the use in comprehensive Earth System Models. The manuscript is very well structured, organized and written. The main part of the manuscript describes in a formal way the physics behind the EVA algorithm which is important for groups that would like to apply the algorithm for their individual models. However, for the general readership the manuscript would benefit including a number of paragraphs introducing general physical concepts in the context of stratospheric volcanic aerosols that are referred to in the body of the manuscript.

I suggest final publication of the manuscript when the minor points below are addressed in the revised version.

**Specific**

*Introduction*

The introduction lacks two or three additional paragraphs for assisting the reader that is not a specialist to get a basic overview about the physical mechanisms involved in the context of stratospheric sulfate. This relates to a general introduction into the basic concepts of the gas-particle conversion in the stratosphere and the general effect of the volcanic sulfate aerosols on the modification of incoming solar radiation (e.g. related to EXT, SSA and ASY).

A second point relates to the availability of volcanic reconstructions carried out in former studies as introduced in chapter #2 (e.g. Crowley and Unterman,2013; Gao et al. (2008)) and uncertainties involved. This could also be used to address different sources and amounts of uncertainties involved in volcanic reconstructions. The authors state in one of their following chapters on the uncertainties involved in their reconstruction approach based on specific assumptions, for instance based on the Pinatubo eruption. However, taken into account the large uncertainties involved in the reconstruction of stratospheric aerosol loadings based on ice core data and according dating uncertainties this might put some of those EVA-specific uncertainties into perspective.

*Existing data set and approaches*

There is one earlier volcanic reconstruction based on Crowley (2000) which was used as forcing by a number of climate simulations. The discussion of the reconstruction of the volcanic forcing already includes most of the mechanisms implemented for the later reconstruction of Crowley and Unterman 2013. Maybe the authors can add this data set and the references as additional source for volcanic reconstructions.

*The EVA approach*

p. 9, l. 19: The authors mention several times the Brewer-Dobson circulation and its significance for the aerosol distribution and transport – A small paragraph in the introduction or within this chapter would help to understand the overall significance and mechanisms controlling the BD circulation.

A second general remark relates to the importance of the specific synoptic meteorological conditions for the individual outbreaks including the timing in the annual cycle (cf. p. 9 27 ff.) – A few words on how this could impact on the eventual depositing of the sulfate in Greenland and Antarctic ice cores could help to motivate the high degree of uncertainty involved in the reconstruction based on ice cores.

*Sample Results*

p. 15 l. 17: Is there a way to improve the accuracy of small-type extratropical high latitudes eruptions, knowing that there impact in EVA might be too exaggerated?

p. 16 l. 3ff: Is there an explanation why the linear scaling used in EVA for Tambora produces lower AOD values than the non-linear scaling in CU13 ?

*Conclusion and outlook*

The EVA algorithm will most likely have also an important role in the PMIP4 simulations and the VolMIP project. A concluding paragraph on the linkages with PMIP4 could also serve as an outlook for applications of EVA in producing consistent volcanic data sets for paleoclimatic modelling of the Holocene and the last 2,000 years.

---

## Referee Comment (RC2) · ANL LeGrande (Referee) · 10 Aug 2016

The EVA product is a much needed addition to doing long-term simulations that include volcanoes. It is an intermediary between having a full-scale (and computationally expensive) aerosol microphysics module and a highly parameterized sulfate aerosol forcing – or worse, alterations in solar constant or similar as a stand-in for actual volcanoes.

This paper is well-written and well-thought out. My main suggestion is how the considerable uncertainty in the measurements of volcanic aod/reff etc. influences the principal results of the EVA product.

Also, N.B., I am not the best person to comment on the radiation tables and methods applied, etc. – I leave it to other reviewers to provide commentary as necessary.

[Figure]

There are quite a few acronyms. It would be useful to have a small table / glossary.

Abstract

The abstract could be changed a small bit to make this more than a summary of the methods: A sentence or two at the top for motivation might illustrate why this is such a useful tool (as in the first paragraph of the intro). A sentence at the end to say its results fall within the spread of sophisticated modules (i.e., as in Zanchettin 2016) would emphasize this.

Intro

I think maybe a short summary of how models approximate volcanoes would be helpful. (And show how this is many-steps better than the altered solar-constant variety of 'volcano'.)

Line 20: the volmip paper is Zanchettin et al 2016, is this what you mean?

Section 3.1

18 Tg for Pinatubo SO2 is totally reasonable – but some use estimates as low as 14 Tg. Can you acknowledge this uncertainty here and address later on.

Depending on when you downloaded the Sato / GISS strataer files, the Reff and AOD may not include the most recent updates to the SAGE II estimates. Probably it is best to ask Makiko directly about when the update was of the data you downloaded. It is important because, for instance, the shape of 91Pinatubo AOD, for instance, is considerably different in [Vernier et al., 2011]. The shape looks more like an exponential increase with slow decay – reaching max AOD much more quickly than in the previous SAGE II interpretations (eg Sato93). This SAGE II does feed into Makiko's updates. The Vernier group redid its Reff estimates, too.

It is likely that the Sato method or CU13 method is capable of getting Reff for eruptions larger than Pinatubo (>0.15aod), and this uncertainty should be explored here as well.

Perhaps taking Reff from some previous full aerosol microphysics simulations? (Which I know has its own problematic spread).

Hemispheric Asymmetry

For the 91Pinatubo case – is the Cerro Hudson contribution considered? (Maybe it helped give extra aod to the SH).

Section 3.6

This uncertainty for large eruptions is important and needs to have more mention up top. It is not clear that a single threshold is the appropriate way to implement the 2/3 power. Can there be a continuous function.

Section 4

I am surprised that the [Carn et al., 2016] paper isn't used in here. It is new, but it is comprehensive, and has emerged as one of the principal papers used for volcanic so2 from measurement.

Carn, S. A., L. Clarisse, and A. J. Prata (2016), Multi-decadal satellite measurements of global volcanic degassing, Journal of Volcanology and Geothermal Research, 311, 99-134,doi:10.1016/j.jvolgeores.2016.01.002.

Vernier, J. P., L. W. Thomason, J. P. Pommereau, A. Bourassa, J. Pelon, A. Garnier, et al. (2011), Major influence of tropical volcanic eruptions on the stratospheric aerosol layer during the last decade, Geophysical Research Letters, 38(12), n/a-n/a,doi:10.1029/2011gl047563.

---

## Author Response (AR1)

Dear Editor Andrea Stenke,

Please find below a copy of our responses to the reviewers comments, as well as a revised version of manuscript with revisions marked.

We are grateful to the reviewers for their helpful comments. The most significant change to the manuscript is the addition of background information, now included at the beginning of Sec. 2. We agree with the reviewers that some background will help many readers appreciate the rest of the material of the manuscript.

Please contact me for any further information or clarification about the revision of the manuscript.

Sincerely,

Matthew Toohey

On behalf of all co-authors

We thank the reviewers for their helpful comments. In the following, reviewer comments are in black, our author responses in blue.

Reviewer 1:

General

The manuscript describes an algorithm for calculating temporal, latitudinal, vertical and wavelength dependent distribution of sulfate aerosols for the use in comprehensive Earth System Models. The manuscript is very well structured, organized and written. The main part of the manuscript describes in a formal way the physics behind the EVA algorithm which is important for groups that would like to apply the algorithm for their individual models. However, for the general readership the manuscript would benefit including a number of paragraphs introducing general physical concepts in the context of stratospheric volcanic aerosols that are referred to in the body of the manuscript.

I suggest final publication of the manuscript when the minor points below are addressed in the revised version.

Specific

Introduction

The introduction lacks two or three additional paragraphs for assisting the reader that is not a specialist to get a basic overview about the physical mechanisms involved in the context of stratospheric sulfate. This relates to a general introduction into the basic concepts of the gas-particle conversion in the stratosphere and the general effect of the volcanic sulfate aerosols on the modification of incoming solar radiation (e.g. related to EXT, SSA and ASY).

We agree, and have added background paragraphs in Sec 2.

A second point relates to the availability of volcanic reconstructions carried out in former studies as introduced in chapter #2 (e.g. Crowley and Unterman,2013; Gao et al. (2008)) and uncertainties involved. This could also be used to address different sources and amounts of uncertainties involved in volcanic reconstructions. The authors state in one of their following chapters on the uncertainties involved in their reconstruction approach based on specific assumptions, for instance based on the Pinatubo eruption. However, taken into account the large uncertainties involved in the reconstruction of stratospheric aerosol loadings based on ice core data and according dating uncertainties this might put some of those EVA-specific uncertainties into perspective.

We appreciate the comment. We do not want to include a lengthy discussion of the different uncertainties involved in ice core reconstructions, since this work focuses specifically on the methods of producing optical properties from given $SO_2$ injection amounts. Nonetheless, we agree that the uncertainty in proxy records is important, and supports our justifications for the construction of a simple forcing (e.g., compared to the use of coupled aerosol models). We include now a mention of ice cores in the Introduction, and have added the following sentence to the Conclusions:

"For most purposes, inaccuracies in forcing due to the simple approach of EVA are likely small compared to uncertainties in our knowledge of the properties of past volcanic eruptions inferred from proxies like ice cores. "

Existing data set and approaches

There is one earlier volcanic reconstruction based on Crowley (2000) which was used as forcing by a number of climate simulations. The discussion of the reconstruction of the volcanic forcing already includes most of the mechanisms implemented for the later reconstruction of Crowley and Unterman 2013. Maybe the authors can add this data set and the references as additional source for volcanic reconstructions.

We now make reference to the Crowley (2000) reconstruction within the Crowley and Unterman (2013) subsection, since the two are very similar.

The EVA approach

p. 9, l. 19: The authors mention several times the Brewer-Dobson circulation and its significance for the aerosol distribution and transport – A small paragraph in the introduction or within this chapter would help to understand the overall significance and mechanisms controlling the BD circulation.

Yes, good point. This has been added with the radiation background material at the beginning of Sec. 2.

A second general remark relates to the importance of the specific synoptic meteorological conditions for the individual outbreaks including the timing in the annual cycle (cf. p. 9 27 ff.) – A few words on how this could impact on the eventual depositing of the sulfate in Greenland and Antarctic ice cores could help to motivate the high degree of uncertainty involved in the reconstruction based on ice cores.

Indeed, the importance of specific synoptic meteorological conditions on the hemispheric asymmetry of the aerosol distribution is discussed and dealt with in Sec 3.6. We have put a pointer to this issue in the description of the seasonal cycle of mixing and transport, and explicitly mentioned synoptic meteorology in Sec 3.6.

Sample Results

p. 15 l. 17: Is there a way to improve the accuracy of small-type extratropical high latitudes eruptions, knowing that there impact in EVA might be too exaggerated?

We have considered this, but to implement variations in loss timescales depending on altitude of injection represents a large increase in complexity of the EVA code. Also, there are presently insufficient observations to constrain how aerosol lifetime depends on eruption magnitude. Perhaps model results can help in this regard, but we prefer to address this in future work.

We have slightly expanded the explanation of the source of the overestimation for extratropical weak eruptions, and note that there are few observations on which to base a relationship of lifetime versus eruption magnitude.

p. 16 l. 3ff: Is there an explanation why the linear scaling used in EVA for Tambora produces lower AOD values than the non-linear scaling in CU13 ?

The reason comes back to the different linear scalings, discussed in Sec 3.7:

"The linear scaling used in EVA is significantly less steep than that of CU13, a result of the lower peak global mean AOD estimate for Pinatubo from CCMI compared to Sato/GISS, and the larger estimate of SO2 injection from satellite sensors compared to the ice core-derived estimate of CU13."

As a result, the linear relationship for EVA lies "below" the non-linear curve from CU13, up to $SO_2$ injections a little larger than Tambora, as shown graphically in Fig 13.

Conclusion and outlook

The EVA algorithm will most likely have also an important role in the PMIP4 simulations and the VolMIP project. A concluding paragraph on the linkages with PMIP4 could also serve as an outlook for applications of EVA in producing consistent volcanic data sets for paleoclimatic modelling of the Holocene and the last 2,000 years.

We have added references to PMIP and VolMIP in the concluding paragraph.

Reviewer 2 (Allegra LeGrande)

The EVA product is a much needed addition to doing long-term simulations that include volcanoes. It is an intermediary between having a full-scale (and computationally expensive) aerosol microphysics module and a highly parameterized sulfate aerosol forcing – or worse, alterations in solar constant or similar as a stand-in for actual volcanoes.

This paper is well-written and well-thought out. My main suggestion is how the considerable uncertainty in the measurements of volcanic aod/reff etc. influences the principal results of the EVA product.

Also, N.B., I am not the best person to comment on the radiation tables and methods applied, etc. – I leave it to other reviewers to provide commentary as necessary.

There are quite a few acronyms. It would be useful to have a small table / glossary.
A table of acronyms has been added in Appendix A.

Abstract

The abstract could be changed a small bit to make this more than a summary of the methods: A sentence or two at the top for motivation might illustrate why this is such a useful tool (as in the first paragraph of the intro). A sentence at the end to say its results fall within the spread of sophisticated modules (i.e., as in Zanchettin 2016) would emphasize this.

We have added a short motivational intro to the abstract as suggested. We have added mention of the result of Zanchettin et al. (2016) in the Conclusions and Outlook: since it is not a result of the present paper, we feel it belongs more naturally there than in the abstract.

Intro

I think maybe a short summary of how models approximate volcanoes would be helpful. (And show how this is many-steps better than the altered solar-constant variety of 'volcano'.)

We have added a short paragraph introducing some of the ways volcanic forcing has historically been implemented in climate models, within Sec. 2.

Line 20: the volmip paper is Zanchettin et al 2016, is this what you mean?

We originally referred to the Zanchettin et al. (2015) PAGES magazine article, specifically to Fig. 1, but since the same material is included in the 2016 GMD paper, we have now changed this reference as suggested.

Section 3.1

18 Tg for Pinatubo SO2 is totally reasonable – but some use estimates as low as 14 Tg. Can you acknowledge this uncertainty here and address later on.

We have added the uncertainty for the Guo et al. (2004) $SO_2$ injection estimate (i.e., $18 \pm 4$ Tg $SO_2$), and noted the corresponding uncertainty in the parameter $A$ later in Sec 3.1.

Depending on when you downloaded the Sato / GISS strataer files, the Reff and AOD may not include the most recent updates to the SAGE II estimates. Probably it is best to ask Makiko directly about when the update was of the data you downloaded. It is important because, for instance, the shape of 91Pinatubo AOD, for instance, is considerably different in [Vernier et al., 2011]. The shape looks more like an exponential increase with slow decay – reaching max AOD much more quickly than in the previous SAGE II interpretations (eg Sato93). This SAGE II does feed into Makiko's updates.

After direct contact with Dr. Sato, we are confident that the Sato et al. (1993, with updates) we have downloaded and used is, for the Pinatubo period, consistent with early SAGE retrievals, and has not been subsequently updated with newer SAGE retrievals. It appears that Dr. Sato has used updated Pinatubo-period forcing data (provided by J.P. Vernier) in recent studies, but the "official" data set, downloadable from the NASA/GISS website has remained static since 2012.

The Vernier group redid its Reff estimates, too.

The SAGE_4λ (or CCMI) dataset contains estimates of the mean aerosol radius ($r_{mean}$). Under the assumption of a single-mode, log-normal distribution, we can translate $r_{mean}$ into $r_{eff}$ (see below). This estimate is quite similar to the older Sato dataset. Nonetheless, we will include the CCMI-based $r_{eff}$ estimate in an updated Fig 1.

[Figure]

It is likely that the Sato method or CU13 method is capable of getting Reff for eruptions larger than Pinatubo (>0.15aod), and this uncertainty should be explored here as well.

It is important that there are no observations, or even indirect evidence of the size distributions of aerosol from eruptions larger than Pinatubo. Hence, there is considerable uncertainty in the effective radius for such eruptions. We have added a few sentences to make this clear, in Sec. 4.2 when discussing the EVA-produced $r_{eff}$ for Tambora.

Perhaps taking Reff from some previous full aerosol microphysics simulations? (Which I know has its own problematic spread).

We plan to compare the EVA results with interactive aerosol model results in future work, but at the moment, the models show a very large spread and don't really help to reduce uncertainty. We include now a reference to Stoffel et al., (2015) who report the maximum $r_{eff}$ produced by their aerosol model in simulations of Tambora.

Hemispheric Asymmetry

For the 91Pinatubo case – is the Cerro Hudson contribution considered? (Maybe it helped give extra aod to the SH).

Estimates of $SO_2$ injection by the Aug 1991 eruption of Cerro Hudson are around 1.5 Tg $SO_2$ (Doiron et al., 1991), i.e., about 1/10[th] that of Pinatubo. Inspection of the SAGE/CCMI data shows clearly that due to its lower stratospheric injection height, the aerosol cloud from Cerro Hudson was short-lived, and its contribution to the maximum post-Pinatubo SH AOD, in around December 1991, was likely negligible. We have added a sentence to be clear that we do not believe Cerro Hudson had a large impact on the satellite AOD record, i.e., that Pinatubo aerosol really was evenly distributed to the two hemispheres.

Section 3.6

This uncertainty for large eruptions is important and needs to have more mention up top. It is not clear that a single threshold is the appropriate way to implement the 2/3 power. Can there be a continuous function.

We agree that from a theoretical perspective, a single function would be satisfying. Obviously, both processes (nucleation of new particles and condensation onto pre-existing particles) can happen at the same time, and the best parameterization would include a smooth variation of the proportion of one process or the other. We hope to investigate this in future research. However, the threshold based approach does have the advantage of being conceptually straight-forward, and is consistent with prior work. Also, since the threshold-based function is smooth and continuous, it seems likely that a more complicated parameterization would likely be practically very similar in form. Therefore, we maintain the use of the threshold-based function, until it can be shown to be deficient.

We have edited and expanded the discussion of the linear-nonlinear threshold approach in Sec 3.7 to reflect some of these issues, including the following text:

"The present scheme retains consistency with the reconstruction of CU13, and has the advantage of simplicity, at least for the majority or eruptions for which AOD is a simple linear scaling of sulfate aerosol mass. Scaling considerations for extremely large eruptions should be understood to be a major source of uncertainty in any volcanic forcing reconstruction."

Section 4

I am surprised that the [Carn et al., 2016] paper isn't used in here. It is new, but it is comprehensive, and has emerged as one of the principal papers used for volcanic so2 from measurement.

We used the Brühl et al (2015) estimates for the most recent eruptions, since these are based on vertically-resolved MIPAS $SO_2$ measurements, and therefore more accurately estimate the $SO_2$ injection above the tropopause than the OMI-based, vertically integrated estimates of Carn et al. (2016). However, we agree that the Carn et al. (2016) data set is very comprehensive, and have added a pointer to it in the conclusions, regarding the potential use of EVA with different $SO_2$ injection data sets.

Carn, S. A., L. Clarisse, and A. J. Prata (2016), Multi-decadal satellite measurements of global volcanic degassing, Journal of Volcanology and Geothermal Research, 311, 99-134,doi:10.1016/j.jvolgeores.2016.01.002.

Vernier, J. P., L. W. Thomason, J. P. Pommereau, A. Bourassa, J. Pelon, A. Garnier, et al. (2011), Major influence of tropical volcanic eruptions on the stratospheric aerosol layer during the last decade, Geophysical Research Letters, 38(12), n/an/a,doi:10.1029/2011gl047563.

Stoffel, M., Khodri, M., Corona, C., Guillet, S., Poulain, V., Bekki, S., Guiot, J., Luckman, B. H., Oppenheimer, C., Lebas, N., Beniston, M. and Masson-Delmotte, V.: Estimates of volcanic-induced cooling in the Northern Hemisphere over the past 1,500 years, Nat. Geosci., advance on, doi:10.1038/ngeo2526, 2015.

Doiron, S. D., Bluth, G. J. S., Schneltzer, C. C., Krueger, A. J. and Walter, L. S.: Transport of the Cerro Hudson SO2 clouds, Eos, Trans. Am. Geophys. Union, 72(45), 489–489, doi:10.1029/90EO00354, 1991.

Zanchettin, D., Timmreck, C., Khodri, M., Robock, A., Rubino, A., Schmidt, A. and Toohey, M.: A coordinated modeling assessment of the climate response to volcanic forcing, Past Glob. Chang. Mag., 23, 54–55, 2015.

**Easy Volcanic Aerosol (EVA v1.0): An idealized forcing generator for climate simulations**

Matthew Toohey[1,2], Bjorn Stevens[1], Hauke Schmidt[1], Claudia Timmreck[1]

[1] Max Planck Institute for Meteorology, Hamburg, 20146, Germany

[2] GEOMAR, Helmholtz Centre for Ocean Research Kiel, 24105, Germany

*Correspondence to*: Matthew Toohey (mtoohey@geomar.de)

**Abstract.** Stratospheric sulfate aerosols from volcanic eruptions have a significant impact on the Earth's climate. To include the effects of volcanic eruptions in climate model simulations,  the Easy Volcanic Aerosol (EVA) forcing generator  provides stratospheric aerosol optical properties as a function of time, latitude, height and wavelength for a given input list of volcanic eruption attributes. EVA is based on a parameterized three-box model of stratospheric transport, and simple scaling relationships used to derive mid-visible (550 nm) aerosol optical depth and aerosol effective radius from stratospheric sulfate mass. Pre-calculated look up tables computed from Mie theory are used to produce wavelength dependent aerosol extinction, single scattering albedo and scattering asymmetry factor values. The structural form of EVA, and the tuning of its parameters, are chosen to produce best agreement with the satellite-based reconstruction of stratospheric aerosol properties following the 1991 Pinatubo eruption, and with prior millennial-time scale forcing reconstructions including the 1815 eruption of Tambora. EVA can be used to produce volcanic forcing for climate models which is based on recent observations and physical understanding, but internally self-consistent over any time-scale of choice. In addition, EVA is constructed so as to allow for easy modification of different aspects of aerosol properties, in order to be used in model experiments to help advance understanding of what aspects of the volcanic aerosol are important for the climate system.

**1 Introduction**

Radiative forcing by variations of stratospheric sulfate aerosol from volcanic eruptions is one of the strongest drivers of natural climate variability (Crowley, 2000; Schurer et al., 2013). To reproduce the radiative forcing of past volcanic eruptions, and thereby the related climate variability, climate model simulations require estimates of the optical properties of volcanic stratospheric aerosols. Prognostic stratospheric aerosol schemes are available, however, such schemes are computationally expensive, and many of the processes underlying them are still not well understood. For these reasons, transient simulations such as historical or millennium simulations usually rely on prescriptive volcanic forcing reconstructions (where "forcing" hereafter refers not specifically to "radiative forcing", but rather to any external driver of climate variability prescribed in climate model simulations). Our knowledge of past eruptions and their climate forcing is based on satellite and ground-based measurements during the recent decades, and longer term histories can be inferred from proxies like ice cores. Different volcanic aerosol forcing sets are currently available, which use different data sources,

different methodologies for combining data sources, and provide different—often incomplete—representations of aerosol properties needed for the radiative calculations of climate models.

The response of the Earth system to volcanic forcing simulated by climate models has been seen to be unrealistic in a number of prior studies. Stratospheric heating, due to the absorption of infrared radiation by volcanic aerosols appears to be overestimated in some models (Driscoll et al., 2012). Tropospheric cooling, while relatively realistically simulated for recent eruptions (Santer et al., 2014), appears to be too strong in model simulations for a number of large past eruptions, most noticeably for the eruptions of Tambora in 1815 (Brohan et al., 2012) and Samalas in 1257 (Stoffel et al., 2015). Post-volcanic anomalies of atmospheric circulation, inferred from observations, are not robustly simulated by models with prescribed forcing, either at the surface (Driscoll et al., 2012), or in the stratosphere (Charlton-Perez et al., 2013). There is also a large degree of inter-model spread in the temperature response to volcanic eruptions, and even in the radiative anomalies created by prescribed volcanic forcing (Zanchettin et al., 2016). Because there are differences in the forcing data sets used, some of which may result from differences in their implementation, it remains unclear to what degree inter-model spread in response to volcanic forcing is attributable to differences in the climate models (i.e., model uncertainty) or differences in forcing (or its implementation).

In general, to isolate model uncertainty in the response to external forcings, it is desirable to have a single forcing implementation strategy, which can be applied consistently in different models. To test the sensitivity to different aspects of the forcing, and gain understanding as to what aspects of the forcing are important for the climate response, it is further desirable to have a forcing strategy which is flexible enough to be used in sensitivity studies. Such motivations inspired the "Easy Aerosol" approach to prescribed tropospheric aerosols (Voigt et al., 2014), wherein the spatial structure of tropospheric aerosols was defined by simple analytical functions in latitude and longitude. Building upon the Easy Aerosol approach, the MACv2-SP module provides relatively realistic representations of anthropogenic aerosol plumes, with a large degree of flexibility and utility for idealized studies (Stevens et al., 2016)(Stevens et al., in preparation).

We present here a description of the Easy Volcanic Aerosol (EVA) forcing generator for use in climate model simulations. EVA provides models with the full optical properties of volcanic aerosols in terms of wavelength dependent aerosol extinction, single scattering albedo, and asymmetry factor, given an input list of eruption locations, dates, and estimated stratospheric sulfur injections. The spatio-temporal structure of the prescribed forcing aims to strike a balance between being realistic (compared to modern observations), and generic, therefore producing consistent representations of eruptions over arbitrary time periods. The underlying parameterization is also readily modifiable, lending itself naturally to idealized sensitivity studies. EVA is comprised of a FORTRAN module that can be called directly by climate models, or can be used offline to produce forcing files which a model reads upon integration.

EVA builds directly upon the methods and results of previous volcanic aerosol forcing reconstructions, which are briefly described in Sec. 2. The EVA approach is detailed in Sec. 3, and a brief comparison with other reconstructions is included in Sec. 4. A summary of EVA, and outlook to potential uses and future versions is included in Sec 5. A list of acronyms is provided in Appendix A.

**2  Volcanic aerosol forcing: theory and practice**

Volcanic eruptions impact climate primarily though the release of sulfur gases, mostly in the form of sulfur dioxide ($SO_2$). In the atmosphere, volcanic $SO_2$ is chemically converted to sulfuric acid ($H_2SO_4$), which forms liquid sulfate aerosol particles (Kremser et al., 2016). Sulfate aerosols in the stratosphere tend to have typical radii of tenths of microns (i.e., 0.1-1.0 µm) (Junge et al., 1961). The size distribution of stratospheric sulfate aerosol is often approximated by the log-normal distribution, described by a distribution mean, and standard deviation. From a radiative standpoint, the area weighted mean radius, or effective radius ($r_e$), is an important property of the size distribution (Hansen and Travis, 1974).

Sulfate aerosols affect the transfer of radiation through the atmosphere. Like gases or other particulate matter, sulfate aerosols can absorb and scatter incoming radiation, with a spectrally dependent signature that depends on the underlying size distribution of the aerosol. Assuming spherical aerosol particles and based on inputs describing the size distribution, and the real and imaginary indices of refraction of the material, Mie theory provides an exact solution of the radiative effects of aerosols, including the proportion of incident radiation absorbed, the proportion scattered, and the variation of scattered light with direction (Hansen and Travis, 1974). There are multiple ways of representing the resulting radiative effects, a common method (e.g., (Stenchikov et al., 1998)) uses the following 3 parameters: aerosol extinction (EXT) represents the total attenuation of incident radiation; the single scattering albedo (SSA) represents the proportion of EXT which is scattered (as opposed to absorbed); and the scattering asymmetry factor (ASY) gives the average cosine of the scattering angle, weighted by the intensity of the scattered light as a function of angle. It has the value 1 for perfect forward scattering, 0 for isotropic scattering and -1 for perfect backscatter. Finally, a very common parameter used to describe aerosol forcing is the aerosol optical depth (AOD), also called the aerosol optical thickness (AOT), which is the integral of the vertical profile of EXT. The unitless AOD therefore describes the amount by which incoming radiation is attenuated through the whole atmosphere.

The gravitational settling velocity of particles of the size of stratospheric aerosol particles is small, therefore the lifetime of sulfate aerosols in the stratosphere is significant (Junge et al., 1961). Lifetime, here, refers to the "e-folding" lifetime—the characteristic timescale of a chemical or physical process in which the time derivative of a quantity is proportional to its amount—defined by the time taken for a decreasing quantity to reach 1/e of its initial amount (Jacob, 1999). Given their relatively long lifetime, the transport of sulfate aerosols is largely controlled by stratospheric dynamics. The strongest winds in the stratosphere are in the zonal (east-west) direction, which act to homogenize stratospheric composition on a fairly fast timescale (Shepherd, 2003). The Brewer-Dobson circulation (BDC) controls the distribution of stratospheric composition in the vertical-meriodional plane (Holton et al., 1995; Shepherd, 2003), and is characterized by seasonally dependent two-way mixing in the midlatitudes, and a slow meridional cell of mass transport characterized by upward motion in the tropics, poleward motion in the midlatitudes and downwelling over the poles.

The radiative effects of volcanic aerosols have been included in climate model simulations using a wide range of methods. The simplest method used involves decreasing the solar constant to reproduce the net global change in surface radiation (Metzner et al., 2012; Yoshimori et al., 2005). This method fails to reproduce the heating of the stratosphere due to aerosol

absorption of infrared radiation, and also negelcts spatio-temporal variation in radiative anomalies. On the other end of the spectrum, interactive stratospheric aerosol models explicitly simulate the evolution of stratospheric aerosols and their radiative impacts. While there is much to be learned from these models, there remain significant uncertainties in the representation and coupling of aerosols, as evidenced by the large inter-model spread in standardized simulations of the 1815 Tambora eruption (Zanchettin et al., 2016). Furthermore, such models are computationally expensive, which usually prohibits their use for long-term (i.e., 50+ yr) simulations.

The most common method of including volcanic effects in climate simulations makes use of "prescribed" volcanic forcing data sets.

 Prior works have used different methods to reconstruct volcanic forcing of climate, and implement these effects in climate model simulations. 
[revised manuscript text omitted]

Input data, specifying the stratospheric sulfur injection properties for a number of volcanic eruptions were collected from a range of sources, summarized in Table 1. The ice core-derived sulfate aerosol loading estimates of Gao et al. (2008) were used to produce $SO_2$ injection estimates for the great Tambora eruption of 1815, as well as the eruptions of Agung (1963), Fuego (1974), and El Chichón (1982). For the 1991 eruption of Pinatubo, we use the estimated total $SO_2$ injection of 18 Tg (9 Tg S) from the Total Ozone Mapping Spectrometer (TOMS) instrument (Guo, 2004). Estimates of stratospheric $SO_2$ injection for a number of relatively smaller eruptions in the 2000's were taken from Brühl et al., (2015), based on the MIPAS $SO_2$ retrievals described by Höpfner et al., (2015).

**3.2 Global mean AOD and effective radius**

Aerosol optical depth in the mid-visible (λ=550 nm) is simulated by EVA making use of the assumption that it can be a simple function of the stratospheric sulfate mass, i.e.:

$$AOD_{550} = f(M_{SO4}) \tag{1}$$

(Stothers, 1984) introduced a simple linear scaling between global sulfate aerosol mass and global mean AOD. Such scalings have a physical foundation (e.g., Charlson et al., 1992) and have long been used to study tropospheric aerosols. A linear relationship between sulfate mass and AOD has been used to convert the sulfate mass estimates of Gao et al., (2007) into radiative properties (Schmidt et al., 2011), and is implicit in the linear scaling of ice core sulfate flux to AOD used by CU13 for eruptions of Pinatubo magnitude and smaller. We apply this same assumption hereafter (up to a threshold $M_*$, see Sec. 3.6), using a scaling factor $A$:

$$AOD_{550} = A\,M_{SO4} \quad \text{(for } M_{SO4} < M_*) \tag{2}$$

Time evolution of sulfate mass is emulated in EVA using a chemical box-model framework. Bluth et al. (1997) introduced a simple single-box model of stratospheric aerosol evolution. Changes in stratospheric sulfate aerosol are controlled by injections of $SO_2$ into the box, subsequent conversion of $SO_2$ into sulfate aerosols, and loss of sulfate aerosols to the troposphere. Describing the production and loss of sulfate mass ($M_{SO4}$, all masses in Tg S) as a function of the $SO_2$ mass ($M_{SO2}$) and characteristic timescales $\tau_{prod}$ and $\tau_{loss}$, respectively, the time tendency of $M_{SO4}$ is:

$$\frac{dM_{SO4}(t)}{dt} = \frac{M_{SO2}(t)}{\tau_{prod}} - \frac{M_{SO4}(t)}{\tau_{loss}} \tag{3}$$

This equation describing the time evolution of $M_{SO4}$ can be solved analytically, for example for a single pulse injection of $M_{SO2}$ at time $t_0$,:

$$M_{SO4}(t) = M_{SO2}(t_0) \left[ 1 - \exp\left(\frac{-t}{\tau_{prod}}\right) \right] \exp\left(\frac{-t}{\tau_{loss}}\right) \tag{4}$$

Equations 2 and 4 can be used to emulate the observed global mean AOD evolution after the Pinatubo eruption. Using the best estimate of a total $SO_2$ injection of 9 Tg S (Guo, 2004), the parameters A, $\tau_{prod}$ and $\tau_{loss}$ can be determined based on comparison with the CCMI global mean AOD anomaly timeseries (Fig. 1). Given the uncertainties in the CCMI AOD in the first months after the Pinatubo eruption, due to the gap in the SAGE II observations due to saturation effects (Thomason and Peter, 2006), we have based our fit on the CCMI AOD beginning in July 1992, when SAGE II retrievals of the full tropical stratosphere resumed. Therefore, the fit is not strongly constrained by the peak of global mean AOD in the CCMI reconstruction, but rather by the shape of the AOD decay. Best fit is achieved with values of $A=0.0364$, $\tau_{prod}= 180$ d, and $\tau_{loss}=330$ d. The resultant value of A is comparable to that suggested by (Stothers, 1984) (0.0267 when converted into units of mass S rather than mass sulfate aerosol) and the stratospheric loss rate ($\tau_{loss}=330$ d) is consistent with the decay rate of AOD after Pinatubo noted by other researchers (e.g., Bluth et al., 1997). The best fit "production" rate, $\tau_{prod}=180$ d, is perhaps surprising, given its discrepancy from the observed timescale of $SO_2$ decay, which is around 30-35 d (Bluth et al., 1992; Read et al., 1993). In their single-box aerosol model, Bluth et al. (1997) noted the resulting lag between peak observed AOD and the peak in modeled $SO_4$ mass when using an $SO_4$ production timescale equal to the observed $SO_2$ decay rate, and proposed that the rate of $SO_4$ aerosol may be limited by other chemical steps other than the destruction of $SO_2$. We contend that global mean AOD may further depend on the timescale of spatial spreading of the aerosol cloud, since the impact of aerosol contained within a horizontally contained vertical column will be diminished due to shielding effects. Whatever the mechanism responsible, $\tau_{prod}$ should be interpreted as an effective production timescale, which incorporates not only the chemical conversion of $SO_2$ to $SO_4$, but other processes which damp the rise in AOD. An important caveat of this construction is that the peak loading of the simulated $SO_4$ mass is significantly less than what would result from a complete conversion of $SO_2$ to $SO_4$ prior to any loss. For this reason, the scaling factor $A$ is larger than what would be deduced if the peak $SO_4$ loading is assumed to be equal to the $SO_2$ injection (in Tg S). Finally, we note that since estimates of the mass of $SO_2$ injected by Pinatubo ($9 \pm 2$ Tg S, from (Guo, 2004)) have an uncertainty of about 25%, the uncertainty in the scaling factor $A$ is at least this large.

[revised manuscript text omitted]
 aerosol between the NH and SH. For example (discounting the likely small and short-lived impact of the August 1991 Cerro Hudson eruption) the June 1991 eruption of Pinatubo produced NH and SH AOD maxima of similar magnitude (Fig. 3)., but others, On the other hand, tropical eruptions like Agung (1963) and El Chichón (1982), are characterized by substantial hemispheric asymmetry in stratospheric aerosol loading. Asymmetries in the radiative forcing, if large enough, may have a detectable climate impact, for instance through their influence on the latitude of the inter-tropical convergence zone and subsequent changes in tropical monsoon patterns (Haywood et al., 2013; Oman et al., 2006).

[revised manuscript text omitted]

25 Including the minor eruptions of 2000-2014 in the EVA input file improves the comparison over these years. This time period is examined in more detail in Fig. 15. The EVA global mean AOD tracks the variability shown in the CCMI database (based in this time period on CALIPSO satellite data) rather well. The EVA global mean AOD response to the high latitude eruptions of Kasatochi (2008) and Sarychev (2009) exceeds that seen in the CCMI data, and appears to persist longer. This is likely a result of the use of a single aerosol loss rate for all eruptions in EVA, based on the observed decay of the aerosol

30 from Pinatubo. It is likely that the processes and related timescales are different for relatively smaller eruptions in the extratropics, but there is presently little understanding of the relationship between eruption strength and resulting aerosol lifetime and there is no consideration of this potential effect in EVA., which would not take into account reduced lifetimes

[revised manuscript text omitted]

For most purposes, inaccuracies in forcing due to the simple approach of EVA are likely small compared to uncertainties in our knowledge of the properties of past volcanic eruptions inferred from proxies like ice cores. Instead of attempting to perfectly reproduce observed aerosol properties, EVA makes it possible to pose the scientific question as to what aspects of the volcanic aerosol can produce a detectable climate response, thereby providing a means for deepening our understanding of the interaction of the stratospheric aerosol and climate. Future updates to EVA are planned, but in keeping with its original motivation, no attempt will be made to represent all aspects of the observations, only those that can be demonstrated to have a detectable influence on the climate response. Updates will be motivated by new and updated observations and, to

the extent they can reliably constrain remaining uncertainties, information from more complex aerosol models. In the meantime, in its present form, EVA is useful for a variety of purposes, including producing forcing for paleo-modelling simulations, e.g., within the Paleo-Modelling Intercomparison Project (PMIP) (Kageyama et al., 2016), and for idealized volcanic forcing experiments such as those within the Model Intercomparison Project on the climate response to

5  Volcanic forcing (VolMIP) (Zanchettin et al., 2016). Additional potential uses of EVA include,  decadal prediction simulations in the case of a major eruption (Timmreck et al., 2016),  filling gaps in satellite-based forcing reconstructions, or  in experiments aiming to assess what aspects of the stratospheric aerosol forcing lead to a detectable climate response.

**Code Availability**

10  EVA version 1.0 code, a user's manual, sample input data files and driver scripts, are included as supplementary material.

**Appendices**

Table A.1: Acronyms

| Acronym | Meaning | Notes |
| --- | --- | --- |
| AOD | Aerosol optical depth | An aerosol optical property (see Sec. 2) |
| AER | Atmospheric and Environmental Research | An aerosol microphysics module (Arfeuille et al., 2014; Weisenstein et al., 1997) |
| ASY | Scattering asymmetry factor | An aerosol optical property (see Sec. 2) |
| BDC | Brewer-Dobson Circulation | The stratospheric dynamics that control the transport and distribution of stratospheric composition |
| CCMI | Chemistry-Climate Model Initiative | Used here as an identifier for the volcanic forcing data set provided for use in CCMI model experiments, based on satellite observations and model results (Eyring and Lamarque, 2013) |
| EVA | Easy Volcanic Aerosol | The volcanic aerosol optical property generator introduced in this work |
| EXT | Aerosol extinction | An aerosol optical property (see Sec. 2) |
| PMIP | The Paleo-Modelling Intercomparison Project | A collaborative research project (Kageyama et al., 2016) |
| SAGE | Stratospheric Aerosol and Gas Experiment | A family of satellite instruments measuring stratospheric aerosol |

| | | |
|---|---|---|
| SSA | Single Scattering Albedo | An aerosol optical property (see Sec. 2) |
| TOMS | Total Ozone Mapping Spectrometer | A satellite instrument (Guo, 2004) |
| VolMIP | The Modelling Intercomparison Project on the climatic response to Volcanic forcing | A collaborative research project (Zanchettin et al., 2016) |

**Acknowledgements**

[revised manuscript text omitted]

Yoshimori, M., Stocker, T. F., Raible, C. C., Renold, M., Yoshimori, M., Stocker, T. F., Raible, C. C. and Renold, M.: Externally Forced and Internal Variability in Ensemble Climate Simulations of the Maunder Minimum, J. Clim., 18(20), 4253–4270, doi:10.1175/JCLI3537.1, 2005.

Zanchettin, D., Khodri, M., Timmreck, C., Toohey, M., Schmidt, A., Gerber, E. P., Hegerl, G., Robock, A., Pausata, F. S. R.,

Ball, W. T., Bauer, S. E., Bekki, S., Dhomse, S. S., LeGrande, A. N., Mann, G. W., Marshall, L., Mills, M., Marchand, M., Niemeier, U., Poulain, V., Rozanov, E., Rubino, A., Stenke, A., Tsigaridis, K. and Tummon, F.: The Model Intercomparison Project on the climatic response toVolcanic forcing (VolMIP): experimental design and forcing input data for CMIP6, Geosci. Model Dev., 9(8), 2701–2719, doi:10.5194/GMD-9-2701-2016, 2016.

[revised manuscript text omitted]